# Practical Adversarial Multivalid Conformal Prediction

**Osbert Bastani**[1], **Varun Gupta**[1], **Christopher Jung**[2],
**Georgy Noarov**[1], **Ramya Ramalingam**[1], **Aaron Roth**[1]
[1] University of Pennsylvania, [2] Stanford University
{obastani, vgup, gnoarov, ramya23, aaroth}@seas.upenn.edu,
csj93@stanford.edu

## Abstract

We give a simple, generic conformal prediction method for sequential prediction that achieves target empirical coverage guarantees on adversarial data. It is computationally lightweight — comparable to split conformal prediction — but does not require having a held-out validation set, and so all data can be used for training models from which to derive a conformal score. Furthermore, it gives stronger than marginal coverage guarantees in two ways. First, it gives *threshold calibrated* prediction sets that have correct empirical coverage even conditional on the threshold used to form the prediction set from the conformal score. Second, the user can specify an arbitrary collection of subsets of the feature space — possibly intersecting — and the coverage guarantees will also hold conditional on membership in each of these subsets. We call our algorithm MVP, short for MultiValid Prediction. We give both theory and an extensive set of empirical evaluations.[1]

## 1 Introduction

Consider the problem of predicting labels $y \in \mathcal{Y}$ given examples $x \in \mathcal{X}$. One popular strategy for expressing uncertainty is to allow the algorithm to produce a *prediction set* $\mathcal{T} \subseteq \mathcal{Y}$ rather than an individual label. We give a simple, practical algorithm for constructing prediction sets in sequential prediction problems over an arbitrary domain $\mathcal{X} \times \mathcal{Y}$, given any data-dependent sequence of *conformal score functions* $s_t : \mathcal{X} \times \mathcal{Y} \to \mathbb{R}_{\geq 0}$. In each round $t$, an example represented by a feature vector $x_t \in \mathcal{X}$ arrives. We can define an arbitrary conformal score $s_t : \mathcal{X} \times \mathcal{Y} \to \mathbb{R}_{\geq 0}$ that can depend on previously observed examples (as well as on $f$) in arbitrary ways. We produce a round-dependent threshold $q_t$, which gives us a prediction set $\mathcal{T}_t = \{y \in \mathcal{Y} : s_t(x_t, y) \leq q_t\}$. We then observe the true label $y_t$; we say our prediction set *covers* $y_t$ if $y_t \in S_t$. Given a coverage target $1 - \delta$, our goal is to produce sets that have correct empirical coverage — i.e. that cover a $1 - \delta$ fraction of the labels (we do not want either over-coverage or under-coverage). We wish to make as few assumptions as possible, so that our method is *robust* to arbitrary and unanticipated distribution shift, and applies to e.g. time series data which are very far from exchangeable. We also want our coverage guarantees to be meaningful not just marginally, but at finer granularities: *conditional* on both the threshold value we choose, and on membership of $x_t \in G$ for a set of groups $G \in \mathcal{G} \subseteq 2^{\mathcal{X}}$ that can be arbitrarily defined and intersecting. Finally, we want our algorithm to have low computational overhead, so that it can be applied as a wrapper on top of arbitrary prediction methods, for both regression and classification. The algorithm we propose achieves these goals and has several desirable properties:

**Worst-Case Empirical Coverage:** Our method has worst case *adversarial* guarantees. The sequence of examples $\{(x_t, y_t)\}_{t=1}^{T}$ needs not be drawn from an exchangeable distribution as it does for standard conformal prediction methods [Shafer and Vovk, 2008] — instead, it can be chosen by an adaptive adversary. The conformal scores $s_t$ can be arbitrary and can depend on data from

---

[1]Code to replicate our experiments can be found at https://github.com/ProgBelarus/MultiValidPrediction.

36th Conference on Neural Information Processing Systems (NeurIPS 2022).

previous rounds (e.g. they can be derived from models trained on all past data, and so there is no need to separate data into training and calibration sets as in split conformal prediction [Lei et al., 2018]). Thus, our method can tolerate time series data, and arbitrary and unanticipated distribution shifts.

**Calibrated, Multivalid Coverage:** Our prediction sets obtain their target empirical coverage level not just marginally, but in a *threshold-calibrated* fashion: for every threshold $q$, they approach $1 - \delta$ coverage on the subsequence of rounds $t$ on which the threshold $q_t = q^2$. We also promise group-conditional coverage. We can specify an arbitrary collection of *groups* $\mathcal{G}$, each group $G \in \mathcal{G}$ representing an arbitrary feature-space subset: $G \subseteq \mathcal{X}$. These groups can intersect in arbitrary ways. For example, $\mathcal{G}$ can be a collection of demographic groups based on race, age, income, or medical history, and each data point $x \in \mathcal{X}$ could represent a member of an arbitrary subset of these groups. Our method promises that simultaneously for every group $G \in \mathcal{G}$, on the subsequence of rounds $t$ for which $x_t \in G$, our intervals obtain the target empirical coverage rate (again, in a calibrated fashion).

**Computationally Lightweight:** Our method is computationally lightweight. At each round $t$, it relies on simple arithmetic calculations involving the historical empirical coverage rates of finitely many candidate thresholds $q_t$ on groups $G$ such that $x_t \in G$. Hence it is comparable in cost to fast split conformal prediction methods [Lei et al., 2018] despite its ability to use all data for model training. We give an implementation of our algorithm and an extensive empirical evaluation. By contrast, prior work by Gupta et al. [2022] that gets comparable theoretical guarantees (in the special case of regression prediction intervals) does not give a practical algorithm: their approach uses the Ellipsoid method with a separation oracle to solve exponentially-sized linear programs at all rounds.

**Nearly Statistically Optimal Rates:** For each threshold $q$ and group $G$, we promise $1 - \delta$ empirical coverage over the rounds $t$ where $x_t \in G$ and the predicted threshold $q_t = q$. Let $n^{G,q}$ denote the total number of such rounds. Even if the labels $y_t$ were drawn from a known distribution and our coverage probability was exactly $1 - \delta$, we would expect our *empirical* coverage on these $n^{G,q}$ rounds to deviate from $1 - \delta$ by a $\pm 1/\sqrt{n^{G,q}}$ term. The prior theoretical coverage guarantees for each $q$ and $G$ of Gupta et al. [2022] differ from their target by $\tilde{O}(\sqrt{T}/n^{G,q})$, which is nearly optimal if $n^{G,q} \geq \Omega(T)$ but sub-optimal otherwise. Our algorithm promises empirical coverage rates for each pair $(G, q)$ that differ from the target by an optimal $\tilde{O}(1/\sqrt{n^{G,q}})$ term.

We give an extensive experimental evaluation of our algorithm in a number of settings, and compare to vanilla split conformal prediction [Lei et al., 2018], as well as prior work that handles limited forms of known distribution shift [Tibshirani et al., 2019], produces conservative groupwise coverage [Foygel Barber et al., 2020], and gives adversarial (but uncalibrated) coverage guarantees [Gibbs and Candes, 2021]. In each setting, we show that our algorithm is competitive with previous work "on their turf" (i.e. in settings for which their assumptions are satisfied and we use their evaluation metrics). We then go on to show that our method gives substantial improvements when either the setting or the evaluation metric becomes more difficult — e.g. when the distribution shift is unanticipated, when we measure group-wise rather than just marginal coverage, or when the data comes in adversarial ordering. In some cases we improve on standard techniques even in standard "benign" settings: for example, we improve on split conformal prediction in an online linear regression setting with iid. data when the evaluation metric is just marginal coverage, but the regression function has to be learned from the same stream of data used to calibrate the prediction intervals. This is because split conformal prediction requires using separate splits of the data for training the regression and calibrating the prediction intervals to maintain exchangeability of the conformal scores — but we are able to use all of the data for both tasks.

---

[2]Calibration is especially important in a distribution free setting, when coverage is measured empirically. If we only asked for the target marginal empirical coverage as Gibbs and Candes [2021] do, rather than for threshold-calibrated prediction sets, one could obtain the right coverage rate by "cheating" in the following uninformative way: predict the full label set $S_t = \mathcal{Y}$ on a $1 - \delta$ fraction of rounds (guaranteeing coverage), and the empty set $S_t = \emptyset$ on the remaining $\delta$ fraction of rounds (guaranteeing miscoverage of the label). This obtains empirical coverage rate $1 - \delta \pm O(1/T)$ marginally, but not conditional on the prediction sets chosen.

## 1.1 Additional Related Work

See Angelopoulos and Bates [2021] for an excellent recent survey of conformal prediction methods. The weaknesses of these methods that we seek to address — namely, that in the worst case they provide only marginal coverage, and that they rely on strong distributional assumptions (typically *exchangeability*) — have been noted before. For example, Romano et al. [2020a] note that marginal coverage guarantees are undesirable and give group conditional guarantees for *disjoint* groups by calibrating separately on each group. This fails when the groups can intersect. Foygel Barber et al. [2020] provide guarantees that are valid conditional on membership in intersecting subgroups $\mathcal{G}$. They take a conservative approach, by computing prediction sets separately for each group and then taking the union of all these sets over the groups of a new individual. As a result, their prediction sets are conservative and do not approach the target coverage level. These results both require exchangeable data. Chernozhukov et al. [2018] obtain approximate marginal coverage guarantees for non-exchangeable time series data coming from a rapidly mixing process. Gendler et al. [2021] study conformal prediction for adversarially perturbed data: they assume that the dataset is drawn exchangeably, but the test examples additionally have their features perturbed by small-norm adversarial noise. Their techniques are different from ours, and leverage the fact that the underlying distribution (except for the perturbations) is exchangeable, which we do not require. Tibshirani et al. [2019] study conformal prediction under *covariate shift*. They adapt conformal techniques to handle the case when both the point of distribution shift and the likelihood ratio between the training and test distribution are known. Gibbs and Candes [2021] give a method that can guarantee target marginal coverage without any assumptions on the data generating process. In contrast, our prediction sets promise not just marginal coverage, but are "threshold-calibrated" and hold also conditional on membership in arbitrary sub-groups. Further notions of conditional coverage different from ours have been studied before in the batch setting, for instance training-, object-, and label- conditional guarantees; see e.g. Vovk [2012], Bian and Barber [2022] for details.

Following a recent resurgence in interest in conformal techniques, a number of papers have proposed conformal scores that have desirable properties [Hoff, 2021, Romano et al., 2019, Angelopoulos et al., 2020, Romano et al., 2020b, Park et al., 2019]. Our work is complementary to this line of work: just like traditional methods of conformal prediction, we too take as input arbitrary conformal scores. Thus we can adopt any of these conformal score functions and inherit their properties, while providing the stronger worst-case guarantees of multivalid coverage.

For the special case of prediction intervals, the type of *multi-valid* prediction that we study was first defined in Jung et al. [2021], who gave a way of obtaining it in the *batch* setting for i.i.d. data, via producing *multicalibrated* estimates of label variances and higher moments. Gupta et al. [2022] proved that there exists an online prediction algorithm that gives the sort of multi-valid prediction intervals that we consider in this work. The algorithm we give in this paper is both much more efficient (theirs was not implementable) and has substantially better (optimal) convergence bounds.

Finally, multivalidity is related to subgroup fairness constraints [Kearns et al., 2018, 2019, Kim et al., 2018], which ask for statistical "fairness" constraints to hold across all subgroups defined by some rich class $\mathcal{G}$. In particular, it is closely related to multicalibration [Hébert-Johnson et al., 2018].

## 2 Setting and Notation

We let $\mathcal{X}$ denote a feature domain and $\mathcal{Y}$ a label domain. $\mathcal{G} \subseteq 2^{\mathcal{X}}$ denotes a collection of subsets of $\mathcal{X}$, which we call "groups". For $x \in \mathcal{X}$, $\mathcal{G}(x) = \{G \in \mathcal{G} : x \in G\}$ is the set of groups that contain $x$. For any integer $T > 0$, $[T] = \{1, \dots, T\}$. The probability simplex over a finite set $A$ is denoted $\Delta A$.

Our online uncertainty quantification is based on a bounded conformal score function $s_t : \mathcal{X} \times \mathcal{Y} \to \mathbb{R}$ which can change in arbitrary ways between rounds $t \in [T]$. Without loss of generality, we assume that the scoring function takes values in the unit interval: $s_t(x, y) \in [0, 1]$ for any $x \in \mathcal{X}, y \in \mathcal{Y}$, and $t \in [T]$. Fix some target coverage rate $1 - \delta$. Ideally, the learner wants to produce prediction sets $\mathcal{T}_t(x_t) = \{y \in \mathcal{Y} : s_t(x_t, y) \leq q_t\}$ that cover the true label $y$ with probability $1 - \delta$ over the randomness of the unknown label distribution: $\Pr_{y|x_t}[y \in \mathcal{T}_t(x_t)] \approx 1 - \delta$. This is equivalent to choosing a conformity threshold $q_t$ such that $\Pr_{y|x_t}[s_t(x_t, y) \leq q_t] \approx 1 - \delta$.

We want to model an adversary that can choose an arbitrary sequence of examples $x_t$ and labels $y_t$. However, because the adversary may choose the label distribution with knowledge of the conformal

score function, we will elide the particulars of the conformal score function and the distribution on labels $y_t$ in our derivation, and instead equivalently imagine the adversary as directly choosing $x_t$ and a distribution over conformal scores $s_t$ conditional on $x_t$ (representing the distribution over conformal scores $s_t(x_t, y_t)$). We may thus without loss view the interaction in the following simplified form:

1. The *adversary* chooses a joint distribution over feature vector $x_t \in \mathcal{X}$ and conformal score $s_t \in [0, 1]$. The learner receives $x_t$ (a realized feature vector), but no information about $s_t$.

2. The learner produces a conformity threshold $q_t$.

3. The learner observes the realized conformal score $s_t$.

As we formally discuss in Definition 3.1 of Section 3 below, the adversary's choice of the joint distribution over $x_t$ and $s_t$ in each round of the above protocol will need to additionally satisfy a mild smoothness property, namely that the score distribution should not be overly concentrated on any subinterval of the $[0, 1]$ range. As we explain there, we can in fact enforce this property algorithmically if necessary, by slightly perturbing the observed scores.

For any round $t \in [T]$, we write $\pi_t = (x_t, s_t, q_t)$ to denote the realized outcomes in round $t$, and $\pi_{t_1:t_2}$ for the *transcript* of the interaction in rounds $t_1 \leq \tau \leq t_2$: $\pi_{t_1:t_2} = ((x_\tau, s_\tau, q_\tau))_{\tau=t_1}^{t_2}$. To denote a concatenation of two transcripts, we use $\oplus$: for example, $\pi_{1:t} = \pi_{1:t-1} \oplus \pi_t$. We write $\Pi^* = (\mathcal{X} \times [0, 1] \times [0, 1])^*$ as the domain of all transcripts. Fixing a learner and an adversary induces a probability distribution over transcripts. Our goal is to derive algorithms with coverage guarantees that hold over the transcript randomness, in the worst-case over all possible adversaries.

Given conformity threshold $q$, we say it *covers* conformal score $s$ if $\mathrm{Cover}(q, s) = 1$, where we define: $\mathrm{Cover}(q, s) = \mathbb{1}[s \leq q]$. To define threshold calibration, we bucket our thresholds using a discretization parameter $m$. For any $m$, we write $B_m(i) = \left[\frac{i-1}{m}, \frac{i}{m}\right)$ for each bucket index $i \in [m-1]$, and $B_m(m) = \left[\frac{m-1}{m}, 1\right]$, so that these buckets evenly partition the unit interval $[0, 1]^3$. For any group $G \in \mathcal{G}$ and bucket $i \in [m]$, we write $G_T(i) = \{t \in [T] : x_t \in G, q_t \in B_m(i)\}$ for the set of rounds in the transcript $\pi_{1:T}$ in which the feature vectors belonged to the group $G$ *and* the chosen threshold $q_t$ was in bucket $i$.

We can now define our main objective in this paper: threshold calibrated multivalid coverage.

**Definition 2.1** (Threshold Calibrated Multivalid Coverage). *Fix a coverage target $1 - \delta$ and a collection of groups $\mathcal{G} \subset 2^{\mathcal{X}}$. Given a transcript $\pi_{1:T}$, a sequence of conformity thresholds $(q_t)_{t=1}^T$ is said to be $(\alpha, m)$-multivalid with respect to $\delta$ and $\mathcal{G}$ for some function $\alpha : \mathbb{N} \to \mathbb{R}$ if:*

$$\left| \frac{1}{|G_T(i)|} \sum_{t \in G_T(i)} (\mathrm{Cover}(q_t, s_t) - (1 - \delta)) \right| \leq \alpha(|G_T(i)|), \qquad \textit{for every } i \in [m] \textit{ and } G \in \mathcal{G}.$$

Note that multivalid coverage is defined by a *function* $\alpha(\cdot)$ of the length of the sequence on which empirical coverage is computed, letting us give fine-grained bounds that scale with the sequence length. In this paper we use the following family of functions, parameterized by a constant $\epsilon > 0$:

$$\alpha(n) = \frac{f(n)}{n}, \quad \text{where} \quad f(n) = \sqrt{(n+1) \log^{1+\epsilon}(n+2)}$$

Here, $f$ is defined so that up to lower-order terms, $\alpha(n) \sim \frac{1}{\sqrt{n}}$; the logarithmic factor that depends on $\epsilon > 0$ serves to ensure the technical condition that the series $\frac{1}{f(n)^2}$ is convergent: $\sum_{n=0}^{\infty} \frac{1}{f(n)^2} = K_\epsilon$, where $K_\epsilon$ is a constant depending only on our choice of $\epsilon$, that will later appear in our bounds.

## 3 Our Algorithm and Analysis

Before we provide the algorithm and its guarantees, we first discuss a needed assumption. Observe that even in the easier setting where the conformal score $s$ is drawn from a fixed, known distribution: $s \sim \mathcal{D}$ — there may not be any threshold $q \in [0, 1]$ that satisfies the desired target coverage value, i.e. that guarantees that $|\mathbb{E}_{s \sim \mathcal{D}}[\mathrm{Cover}(q, s) - (1 - \delta)]|$ is small. Consider for example a distribution that places all its mass on a single value $s$. Then any threshold $q$ covers the $s$ with probability 1 or 0,

---

[3]We can handle non-uniform discretizations of the unit interval as well, with no additional complications.

which for $\delta \notin \{0, 1\}$ is bounded away from our target coverage probability. One could randomize the threshold to get the target marginal coverage rate, but this corresponds to the "cheating" strategy we outline in Footnote 2, and in particular would not satisfy our notion of *threshold calibrated* coverage. Of course, if achieving the target coverage is impossible in the easier distributional setting, then it is also impossible in the more challenging online adversarial setting.

With this in mind, just as with many other approaches to conformal prediction that aim to converge to the correct coverage rate (rather than conservatively over-cover), we will need to assume that our target distributions are not too concentrated on any single point. Following Gupta et al. [2022], we define a class of smooth distributions for which achieving (approximately) the target coverage is always possible for some threshold $q$ defined over an appropriately finely discretized range. Our smoothness condition makes sense even for discrete distributions, so we do not need to assume continuity. To denote the uniform grid on $[0, 1]$, we write $\mathcal{P}^{rm} = \left\{0, \frac{1}{rm}, \frac{2}{rm}, \ldots, 1\right\}$.

We show that we can achieve (approximately) our target coverage goals in the online adversarial setting when the adversary is constrained to playing smooth distributions, which are distributions that do not put too much probability mass on any sufficiently small sub-interval.

**Definition 3.1.** *A distribution $Q \in \Delta([0, 1])$ is $(\rho, rm)$-smooth if*

$$\Pr_{s \sim Q}[s \in [a, b]] \leq \rho \quad \text{for any subinterval } [a, b] \subseteq [0, 1] \text{ of length} \leq \frac{1}{rm},$$

*A joint distribution $\mathcal{D} \in \Delta(\mathcal{X} \times [0, 1])$ is $(\rho, rm)$-smooth if for every $x \in \mathcal{X}$, the marginal conformal score conditional on $x$, $\mathcal{D}|_x$, is $(\rho, rm)$-smooth. An adversary is $(\rho, rm)$-smooth if the joint distribution over $(x_t, s_t)$ is $(\rho, rm)$-smooth at every round $t \in [T]$.*

**Remark 3.1.** *For any $\rho$, the assumption of $(\rho, rm)$-smoothness becomes more mild as $r \to \infty$. For us, $r$ will be a nuisance parameter that we can take as large as we want — we will not have to pay for it either in our running time or our coverage bounds. We can also algorithmically enforce smoothness by perturbing the conformal scores with small amounts of noise from any continuous distribution, and so we should think of smoothness as a mild assumption. Our experiments bear this out.*

We now present the algorithm (MVP — MultiValid Predictor). It resembles the algorithm for online mean multicalibration given in Gupta et al. [2022], which in turn is a multi-group generalization of the "almost deterministic" calibration algorithm of Foster and Hart [2021].

---

**Algorithm 1:** MVP($\delta, \eta, m, r$)

**for** $t = 1, \ldots, T$ **do**

Take as input an arbitrary conformal score $s_t : \mathcal{X} \times \mathcal{Y} \to [0, 1]$.

Observe $x_t$, and for each $i \in [m]$ and $G \in \mathcal{G}(x_t)$, compute:

$$n_{t-1}^{G,i} = \sum_{\tau=1}^{t-1} \mathbb{1}[q_\tau \in B_m(i), \ x_\tau \in G] \qquad \text{Definition 3.2}$$

$$V_{t-1}^{G,i} = \sum_{\tau=1}^{t-1} \mathbb{1}[x_\tau \in G, q_\tau \in B_m(i)] \cdot (\text{Cover}(q_\tau, s_\tau) - (1 - \delta)) \qquad \text{Definition 3.3}$$

$$C_{t-1}^i(x_t) = \sum_{G \in \mathcal{G}(x_t)} \frac{1}{f(n_{t-1}^{G,i})} \left( \exp\left( \eta \frac{V_{t-1}^{G,i}}{f(n_{t-1}^{G,i})} \right) - \exp\left( -\eta \frac{V_{t-1}^{G,i}}{f(n_{t-1}^{G,i})} \right) \right). \qquad \text{From Lemma 3.1}$$

**if** $C_{t-1}^i(x_t) > 0$ for all $i \in [m]$ **then**

Choose threshold $q_t = 0$.

**else if** $C_{t-1}^i(x_t) < 0$ for all $i \in [m]$ **then**

Choose threshold $q_t = 1$.

**else**

Find $i^* \in [m-1]$ such that $C_{t-1}^{i^*}(x_t) \cdot C_{t-1}^{i^*+1}(x_t) \leq 0$.

Define $0 \leq p_t \leq 1$ as follows:[a] $p_t = \left| C_{t-1}^{i^*+1}(x_t) \right| / \left( \left| C_{t-1}^{i^*+1}(x_t) \right| + \left| C_{t-1}^{i^*}(x_t) \right| \right)$.

Choose threshold $q_t = \frac{i^*}{m} - \frac{1}{rm}$ with probability $p_t$ and $q_t = \frac{i^*}{m}$ with probability $1 - p_t$.

Output prediction set $\mathcal{T}_t(x_t) = \{y \in \mathcal{Y} : s_t(x_t, y) \leq q_t\}$.

---

[a] Using the convention that $0/0 = 0$.

First, we give a brief intuitive description of the inner workings of the algorithm. Note that for all groups $G$ and buckets $i$, MVP maintains historical over- (equivalently, under-) coverage amounts over those rounds when the context was in group $G$ (that is, $x_t \in G$), and the threshold played was in bucket $i$ (that is, $q_t \in B_m(i)$). In each round, MVP is confronted with a new context $x_t$, and needs to decide which bucket $i$ to choose the threshold from. For every candidate bucket $i \in [m]$, MVP performs a certain type of normalized softmax aggregation of that bucket's historical over- or undercoverage over all groups $G$ that $x_t$ belongs to. The result is a single signed quantity $C_{t-1}^i$ for each bucket $i$, which summarizes its past coverage performance over all relevant groups $G \ni x_t$: intuitively, if thresholds from bucket $i$ have been mostly overcovering on the relevant groups in the past, we can expect $C_{t-1}^i > 0$, and in the opposite case of significant undercoverage, we can expect $C_{t-1}^i < 0$. The algorithm then simply finds, if such exist, any two adjacent buckets one of which has historically over-covered and the other — under-covered, and randomizes between the two to output a balanced threshold that, it is hoped, neither over- nor under-covers.

Now, we are ready to present our main result — the multivalid coverage guarantees for MVP. The proof of this statement is laid out in the following Section 3.1.

**Theorem 3.1.** *Against any $(\rho, rm)$-smooth adversary and for any adaptively chosen sequence of conformal scores $s_t$, MVP$(\eta, m, r)$ with learning rate $\eta = \sqrt{\frac{\ln(|\mathcal{G}|m)}{2K_\epsilon |\mathcal{G}|m}}$ (Algorithm 1) produces a sequence of thresholds $(q_t)_{t=1}^T$ that is $\left( \left( \sqrt{4K_\epsilon |\mathcal{G}| m \ln(|\mathcal{G}|m)} + \rho T \right) \alpha(\cdot), m \right)$-multivalid in expectation over the randomness of $\pi_{1:T}$ with respect to $\delta$ and $\mathcal{G}$.*

*Consequently, we have for any small $\epsilon > 0$ of our choice (with $K_\epsilon$ a constant that depends only on $\epsilon$):*

$$\mathbb{E}_{\pi_{1:T}} \left[ \max_{G \in \mathcal{G}, i \in [m]} \frac{\left| \sum_{t \in G_T(i)} \left( \mathrm{Cover}\left( q_t, s_t \right) - (1-\delta) \right) \right|}{\sqrt{(|G_T(i)| + 1) \log^{1+\epsilon}(|G_T(i)| + 2)}} \right] \leq \sqrt{4K_\epsilon |\mathcal{G}| m \ln(|\mathcal{G}|m)} + \rho T.$$

**Remark 3.2.** *Since we can take $r$ to be arbitrarily large, for any continuous distribution we can drive the $\rho T$ term to zero. Thus this bound establishes nearly statistically optimal convergence rates for constant $|\mathcal{G}|$ and $m$. Using a simpler analysis analogous to that of Gupta et al. [2022] for mean multicalibration, it is also possible to establish $(\alpha, m)$-multivalidity with $\alpha(n) = O(\sqrt{T \log(|\mathcal{G}|m)}/n + \rho)$, which is optimal in $|\mathcal{G}|$ and $m$ but has a bad dependence on $T$. We believe that our sub-optimal dependence on $|\mathcal{G}|$ and $m$ is an artifact of our analysis, and not a property of our algorithm.*

## 3.1 Analysis

Omitted proofs from this section are presented in full detail in Appendix A. Our analysis here can be seen as an extension of the surrogate loss argument developed in Gupta et al. [2022] for the problem of mean multicalibration. There are two main novel insights that lead to our algorithm and analysis for multivalid coverage. First, Gupta et al. [2022] were unable to extend their simple multicalibration algorithm to prediction interval multivalidity (and instead analyzed an impractical Ellipsoid-based algorithm). Informally this is because they parameterized prediction intervals with two parameters (the lower and upper endpoint), which eliminated the simple one-dimensional structure they were able to exploit for mean multicalibration. In contrast, our prediction intervals are parameterized by a single parameter $q$, which allows us to exploit a simple one-dimensional structure.

Second, the bounds in Gupta et al. [2022] uniformly bound the coverage error for each group $G \in \mathcal{G}$ and bucket $i \in [m]$ by $\tilde{O}(\sqrt{T})$, which is optimal only for subsequences that have $n_T^{G,i} = \Omega(T)$. In contrast, we obtain non-uniform bounds that depend on $n_T^{G,i}$ but not on $T$, and (at least for constant $m$ and $|\mathcal{G}|$) have the optimal $\sqrt{n_T^{G,i}}$ dependence. We achieve this by analyzing a modified surrogate loss, leading to a significant amount of added complexity which accounts for the bulk of our argument.

*Proof sketch for Theorem 3.1.* For each group $G \in \mathcal{G}$, bucket $i \in [m]$, round $t \in [T]$, we would like to bound the coverage error on the subsequence of rounds $\tau$ in which $x_\tau \in G$ and $q_\tau \in B_m(i)$ in terms of the length of that sequence. We give the following notation for these sequence lengths:

**Definition 3.2** (Group-bucket size). *Given a transcript $\pi_{1:t} = ((x_\tau, s_\tau, q_\tau))_{\tau=1}^t$, we define the size for a group $G \in \mathcal{G}$ and a bucket $i \in [m]$ at time $t$ to be:* $n_t^{G,i}(\pi_{1:t}) = \sum_{\tau=1}^t \mathbb{1}[q_\tau \in B_m(i), \ x_\tau \in G]$.

Similarly, for each $G \in \mathcal{G}$, $i \in [m]$ and time $t \in [T]$, we can define the (unnormalized) coverage error on the sequence corresponding to rounds $\tau \le t$ such that $x_\tau \in G$ and $q_\tau \in [m]$:

**Definition 3.3.** *Given transcript $\pi_{1:t}$, the coverage error for group $G \in \mathcal{G}$ and bucket $i \in [m]$ at time $t$ is given as:* $V_t^{G,i} = \sum_{\tau=1}^t \mathbb{1}[x_\tau \in G, q_\tau \in B_m(i)] \cdot v_\delta(q_\tau, s_\tau)$, *where* $v_\delta(q, s) = \mathrm{Cover}(q, s) - (1 - \delta)$.

Note that $V_t^{G,i}$ just records the deviation of the empirical coverage from its target $(1 - \delta)$ on the subsequence of rounds $\tau$ in which $x_\tau \in G$ and $q_t \in B_m(i)$: it takes a positive value if we have *over-covered* on this subsequence and a negative value if we have under-covered.

**Observation 3.1.** *Fix a transcript $\pi_{1:T}$. If for all $G \in \mathcal{G}$ and $i \in [m]$, we have for some constant $c$:* $\left| V_T^{G,i} \right| \le cf(n_T^{G,i})$, *then the resulting sequence of thresholds $(q_t)_{t=1}^T$ is $(c\alpha(\cdot), m)$-multivalid with respect to $\delta$ and $\mathcal{G}$.*

To bound the maximum of our normalized absolute coverage errors across all groups and buckets, i.e. $\max_{G \in \mathcal{G}, i \in [m]} \frac{|V_T^{G,i}|}{f(n_T^{G,i})}$, we use the following surrogate loss:

**Definition 3.4** (Surrogate loss). *Fix a transcript $\pi_{1:t} \in \Pi^*$ and a parameter $\eta \in (0, 1/2)$. Define a surrogate coverage loss function at day $t$ for bucket $i \in [n]$ and group $G \in \mathcal{G}$ as*

$$L_t^{G,i}(\pi_{1:t}) = \exp\left(\eta \frac{V_t^{G,i}}{f(n_s^{G,i})}\right) + \exp\left(-\eta \frac{V_t^{G,i}}{f(n_s^{G,i})}\right),$$

*where $V_t^{G,i}$ are implicitly functions of $\pi_{1:t}$. The overall surrogate coverage loss function is defined as $L_t(\pi_{1:t}) = \sum_{G \in \mathcal{G}, i \in [m]} L_t^{G,i}(\pi_{1:t})$.*

We first show that the increase in the surrogate loss can be bounded in the following way:

**Lemma 3.1.** *Fix $\eta \in (0, \frac{1}{2})$ and a transcript $\pi_{1:t-1}$ for some $t$. For any $\pi_t = (q_t, x_t, s_t)$, we have*

$$L_t(\pi_{1:t-1} \oplus \pi_t) - L_{t-1}(\pi_{1:t-1}) \le \sum_{(G,i) \in A_t(\pi_t)} \eta v_\delta(q_t, s_t) C_{t-1}^{G,i} + \frac{2\eta^2}{f(n_t^{G,i})^2} L_{t-1}^{G,i}(\pi_{t-1}),$$

*where $A_t(\pi_t) = \{(G,i) : G \in \mathcal{G}(x_t), q_t \in B_m(i)\}$ contains $(G,i)$ pairs "active" at time $t \in [T]$, and*

$$C_t^{G,i} = \frac{1}{f(n_t^{G,i})} \left(\exp\left(\eta \frac{V_t^{G,i}}{f(n_t^{G,i})}\right) - \exp\left(-\eta \frac{V_t^{G,i}}{f(n_t^{G,i})}\right)\right).$$

Now, we show that Algorithm 1 guarantees $\sum_{(G,i) \in A_t(\pi_t)} v_\delta(q_t, s_t) C_{t-1}^{G,i}$ is small in expectation over the randomness of the algorithm.

**Lemma 3.2.** *Fix any $t \in [T]$, $\eta \in (0, \frac{1}{2})$, transcript $\pi_{1:t-1}$ recording a realization for the first $t - 1$ rounds and $x_t$. At round $t$, Algorithm 1 chooses a distribution over $q_t$ such that against any $(\rho, rm)$-smooth distribution over conformal scores $s_t$, we have:*

$$\mathbb{E}_{(s_t, q_t)} \left[ \left| \sum_{(G,i) \in A_t(\pi_t)} v_\delta(q_t, s_t) C_{t-1}^{G,i} \right| \pi_{1:t-1} \right] \le \rho L_{t-1}.$$

Carefully telescoping the bounded increase in surrogate loss over each round via Lemma 3.1 and 3.2 and noticing that $1/f(n)^2$ forms a convergent series yields the result of Theorem 3.1. $\qquad\square$

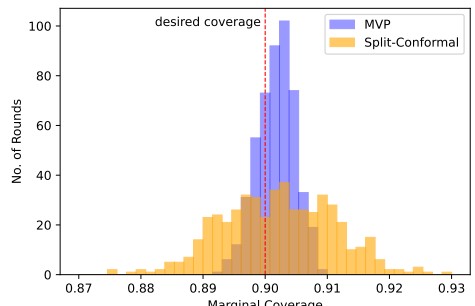 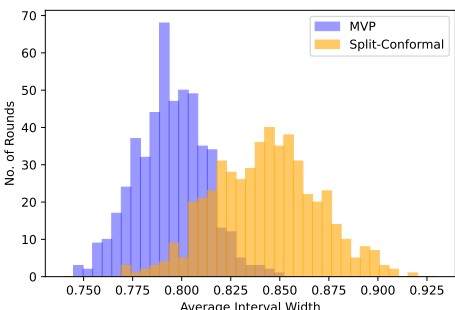

Figure 1: Left: histogram of empirical marginal coverage of `MVP` vs split conformal over 500 repeated trials. Right: histogram of average interval widths. `MVP` gets both empirical coverage that is more tightly concentrated around target (0.9), and narrower coverage interval widths.

## 4 Experiments

In this section, we summarize our experiments[4] — full details can be found in Appendix B. We evaluate `MVP` and compare it to more traditional conformal prediction methods on a variety of tasks. In each case, for a fair comparison, both `MVP` and the compared-to method receive as input the same predictive model and conformal scores.

**Exchangeable data** First, in Section B.1 we study a synthetic linear regression problem in a simple exchangeable (iid.) setting, and compare to split conformal prediction [Lei et al., 2018]. Given a regression model $f_t(x)$, we use conformal score $s_t(x, y) = |f_t(x) - y|$. The regression function is unknown at the outset, and must be learned during the interaction. As we show (Figure 1), `MVP` improves over split conformal prediction even in terms of marginal empirical coverage. This is because to keep the conformal scores exchangeable, split conformal prediction must split the data into two sets: one for training the regression model and one for calibrating prediction sets.[5] In contrast, our method does not require exchangeability, so we can both train the regression model and calibrate our prediction sets on the entire dataset. We are thus able to make better use of the data, and our regression function (and hence our conformal score function) becomes more informative faster. Then, we modify our regression problem so that there are 20 overlapping sub-populations, one of which (consisting of half of the data) has higher label noise. We measure groupwise coverage for `MVP`, for naive split conformal prediction with no knowledge of to-be-covered groups, and the method of Foygel Barber et al. [2020] which guarantees (conservative) groupwise coverage for intersecting groups. We find (Figure 2) that `MVP` significantly improves on both methods, obtaining correct coverage on all groups and (correctly) obtaining much smaller interval width on the low noise group.

**Covariate shift** Next, in Section B.2 we study a regression problem in the presence of covariate shift. First we replicate an experiment of Tibshirani et al. [2019], in which a synthetic covariate shift (with known propensity scores and known changepoint) is simulated on a UCI dataset. The method of Tibshirani et al. [2019] reweighs the calibration set using the propensity scores. `MVP` can also take advantage of propensity scores warm-start `MVP` on the same portion of the dataset that split conformal prediction uses for calibration, sampled with replacement after being re-weighted by the propensity scores. Both algorithms are then evaluated on the shifted distribution. We find both algorithms perform comparably. We then experiment with unanticipated covariate shift simulated on datasets derived from 2018 U.S. Census data provided from the Folktables package [Ding et al., 2021]. We use the quantile-regression based conformal score proposed by Romano et al. [2019] for both methods. We compare to split conformal prediction calibrated on California data and evaluated on the Pennsylvania data. Similarly, we warm-start `MVP` on California data and measure its performance on the Pennsylvania data, finding that `MVP` gets the correct coverage rate and smaller interval widths compared to the split conformal method despite having no knowledge of the distribution shift.

---

[4]Our experiments are fairly lightweight, and can be run e.g. on a standard 12gb RAM Google Colab account.

[5]This is not only a theoretical requirement — split conformal prediction fails badly otherwise.

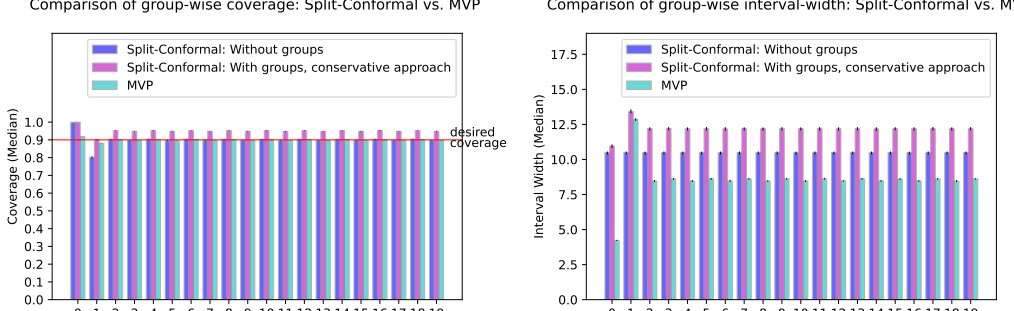

Figure 2: Left: median (over 100 indep. trials) per-group coverage of `MVP` vs split conformal. Right: median group-conditional interval widths. Compared to split conformal, `MVP` obtains target coverage on each group, and narrower interval widths.

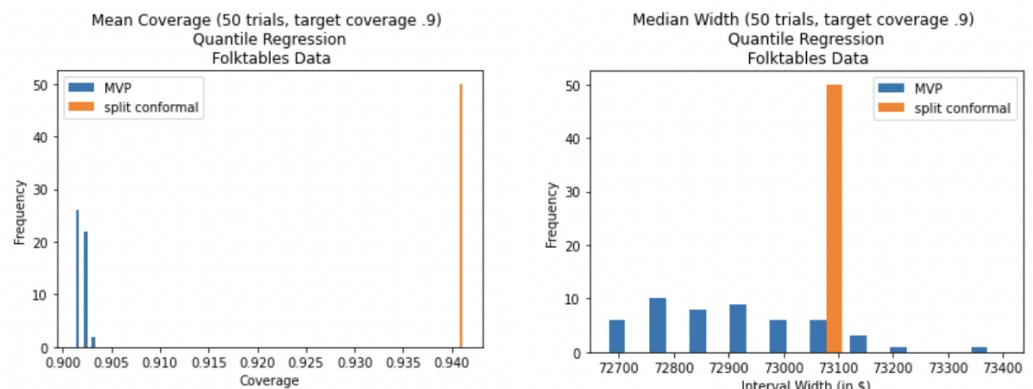

Figure 3: Both `MVP` and split conformal prediction are initialized on California data and evaluated on Pennsylvania data. On the left-hand is a histogram of the coverage for `MVP` and split conformal prediction over 50 trials; the right-hand figure is a histogram of the prediction interval width.

**Time series**   In Section B.3 we evaluate `MVP` on time series data — 20 years of stock returns, in a volatility prediction task. We compare `MVP` to the Adaptive Conformal Inference (ACI) method of Gibbs and Candes [2021], which guarantees marginal (but not threshold- or group- calibrated) coverage for adaptively chosen data. When evaluated in terms of marginal coverage, we find that `MVP` and ACI perform comparably, with ACI's coverage slightly closer to the target, but with `MVP`'s sequence of predicted thresholds exhibiting more stability (Figure B.6). We then complicate the experiment to exhibit the two advantages of `MVP` (groupwise and threshold-calibrated coverage). First, we define 20 intersecting groups of trading days: a period-1 sequence (all the days), a period-2 sequence (the even days), ..., a period-20 sequence (every 20th day). We add perturbations to the stock returns, such that each day receives an amount of noise commensurate with how many groups it belongs to. The point of adding these perturbations to the stock returns on different subsets of the days is to produce a dataset on which the uncertainty of the model is quantifiably different within different groups — thus making it nontrivial to obtain valid coverage on each of those groups by only enforcing valid marginal coverage. Indeed, we find that `MVP` obtains the correct group-wise coverage, whereas ACI fails to: as shown in Figure 4a, ACI undercovers on most groups.[6]

As our next experiment, we produce a fully adversarial sequence by presenting examples to ACI and `MVP` not in time order but in *sorted order by their conformal scores* (see Figure 4b). By construction, this sequence would cause split conformal prediction methods to have 0 coverage, while both ACI and `MVP` guarantee correct marginal coverage on it. However, we find (see Figure B.8d) that ACI achieves its correct marginal coverage by violating threshold calibration: namely, by predicting

---

[6]To balance out this undercoverage, ACI strongly overcovers on points belonging to none of groups 2..20.

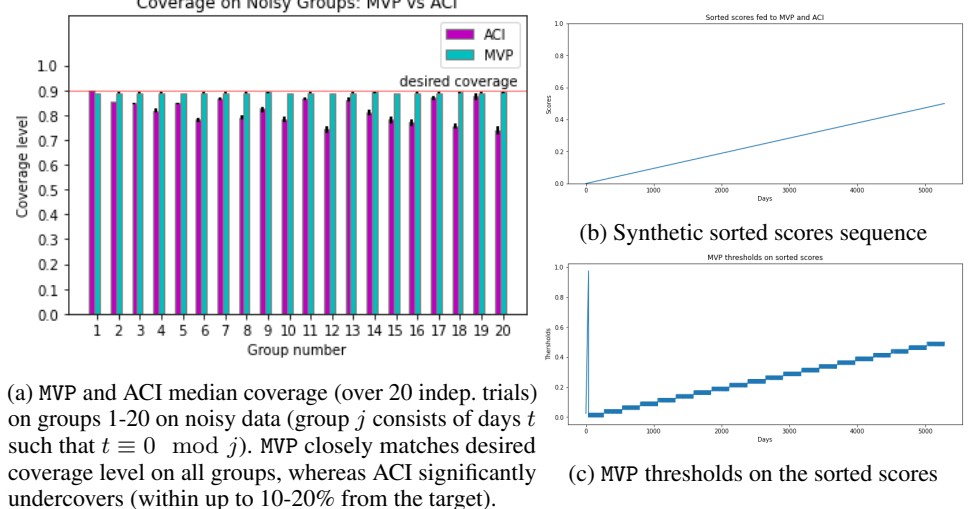

(a) MVP and ACI median coverage (over 20 indep. trials) on groups 1-20 on noisy data (group $j$ consists of days $t$ such that $t \equiv 0 \mod j$). MVP closely matches desired coverage level on all groups, whereas ACI significantly undercovers (within up to 10-20% from the target).

(b) Synthetic sorted scores sequence

(c) MVP thresholds on the sorted scores

Figure 4: MVP vs ACI comparison on time-series data

predominantly the trivial coverage interval (all of $[0, 1]$) and occasionally — short, under-covering intervals.[7] This strategy both yields thresholds that are uninformative about the input sequence of scores, and produces close-to-maximum average interval widths regardless of the actual magnitude of the input scores. In contrast, MVP, by virtue of its threshold calibration, outputs coverage thresholds that correctly track the input sequence of conformal scores (Figure 4c), and hence produces prediction intervals with the correct widths.

**Classification: ImageNet** Finally, in Section B.4 we compare MVP to the work of Angelopoulos et al. [2020] on a large-scale ImageNet classification task. We find that MVP obtains comparable coverage rates and prediction set sizes, despite the fact that the setting is favorable to Angelopoulos et al. [2020] — i.e. the data is i.i.d. and we measure only marginal coverage.

**Potential societal impact of our work:** When the underlying dataset consists of e.g. labeled individuals, our conformal prediction approach achieves target coverage guarantees not only marginally, but on arbitrary collections of user-specified population groups. Thus, we believe that when used responsibly, our method can serve as a key tool for advancing the interests of protected subpopulations, resulting in significant added positive societal impact.

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
