# Practical Adversarial Multivalid Conformal Prediction Supplementary Material

**Osbert Bastani**[1]**, Varun Gupta**[1]**, Christopher Jung**[2]**,**
**Georgy Noarov**[1]**, Ramya Ramalingam**[1]**, Aaron Roth**[1]
[1] University of Pennsylvania, [2] Stanford University
{obastani, vgup, gnoarov, ramya23, aaroth}@seas.upenn.edu,
csj93@stanford.edu

## A    Proofs from Subsection 3.1

### A.1    Proof of Lemma 3.1

*Proof.* Fix $\pi_t = (x_t, s_t, q_t)$. For simplicity, we write $L_t = L_t(\pi_{1:t-1} \oplus \pi_t)$ and $L_t^{G,i} = L_t^{G,i}(\pi_{1:t-1} \oplus \pi_t)$ in the remainder of this proof. For any $(G, i) \notin A_t(\pi_t)$, we have that

$$V_t^{G,i} = V_{t-1}^{G,i}$$
$$n_t^{G,i} = n_{t-1}^{G,i}.$$

and hence $L_t^{G,i} = L_{t-1}^{G,i}$.

On the other hand, for any $(G, i) \in A_t(\pi_t)$, we have

$$V_t^{G,i} = V_{t-1}^{G,i} + v_\delta(q_t, s_t)$$
$$n_t^{G,i} = n_{t-1}^{G,i} + 1.$$

Then, we can bound the change in loss for that group-bucket pair $(G, i) \in A_t(\pi_t)$ as follows:

$$
\begin{aligned}
&L_t^{G,i} - L_{t-1}^{G,i} \\
&= \left( \exp\left( \eta \frac{V_t^{G,i}}{f(n_t^{G,i})} \right) + \exp\left( -\eta \frac{V_t^{G,i}}{f(n_t^{G,i})} \right) \right) - \left( \exp\left( \eta \frac{V_{t-1}^{G,i}}{f(n_{t-1}^{G,i})} \right) + \exp\left( -\eta \frac{V_{t-1}^{G,i}}{f(n_{t-1}^{G,i})} \right) \right) \\
&= \left( \exp\left( \eta \frac{V_{t-1}^{G,i} + v_\delta(q_t, s_t)}{f(n_{t-1}^{G,i} + 1)} \right) + \exp\left( -\eta \frac{V_{t-1}^{G,i} + v_\delta(q_t, s_t)}{f(n_{t-1}^{G,i} + 1)} \right) \right) - \\
&\qquad \left( \exp\left( \eta \frac{V_{t-1}^{G,i}}{f(n_{t-1}^{G,i})} \right) + \exp\left( -\eta \frac{V_{t-1}^{G,i}}{f(n_{t-1}^{G,i})} \right) \right) \\
&\overset{(1)}{\leq} \left( \exp\left( \eta \frac{V_{t-1}^{G,i} + v_\delta(q_t, s_t)}{f(n_{t-1}^{G,i})} \right) + \exp\left( -\eta \frac{V_{t-1}^{G,i} + v_\delta(q_t, s_t)}{f(n_{t-1}^{G,i})} \right) \right) - \\
&\qquad \left( \exp\left( \eta \frac{V_{t-1}^{G,i}}{f(n_{t-1}^{G,i})} \right) + \exp\left( -\eta \frac{V_{t-1}^{G,i}}{f(n_{t-1}^{G,i})} \right) \right) \\
&= \exp\left( \eta \frac{V_{t-1}^{G,i}}{f(n_{t-1}^{G,i})} \right) \left( \exp\left( \eta \cdot \frac{v_\delta(q_t, s_t)}{f(n_{t-1}^{G,i})} \right) - 1 \right) + \exp\left( -\eta \frac{V_{t-1}^{G,i}}{f(n_{t-1}^{G,i})} \right) \left( \exp\left( -\eta \cdot \frac{v_\delta(q_t, s_t)}{f(n_{t-1}^{G,i})} \right) - 1 \right)
\end{aligned}
$$

36th Conference on Neural Information Processing Systems (NeurIPS 2022).

$$\overset{(2)}{\leq} \exp\left(\eta\frac{V_{t-1}^{G,i}}{f(n_{t-1}^{G,i})}\right)\left(\eta\cdot\frac{v_\delta(q_t,s_t)}{f(n_{t-1}^{G,i})}+\frac{2\eta^2}{f(n_{t-1}^{G,i})^2}\right)+\exp\left(-\eta\frac{V_{t-1}^{G,i}}{f(n_{t-1}^{G,i})}\right)\left(-\eta\cdot\frac{v_\delta(q_t,s_t)}{f(n_{t-1}^{G,i})}+\frac{2\eta^2}{f(n_{t-1}^{G,i})^2}\right)$$

$$=\eta\cdot\frac{v_\delta(q_t,s_t)}{f(n_{t-1}^{G,i})}\left(\exp\left(\eta\frac{V_{t-1}^{G,i}}{f(n_{t-1}^{G,i})}\right)-\exp\left(-\eta\frac{V_{t-1}^{G,i}}{f(n_{t-1}^{G,i})}\right)\right)+$$

$$\frac{2\eta^2}{f(n_{t-1}^{G,i})^2}\left(\exp\left(\eta\frac{V_{t-1}^{G,i}}{f(n_{t-1}^{G,i})}\right)+\exp\left(-\eta\frac{V_{t-1}^{G,i}}{f(n_{t-1}^{G,i})}\right)\right)$$

$$=\eta v_\delta(q_t,s_t)C_{t-1}^{G,i}+\frac{2\eta^2}{f(n_{t-1}^{G,i})^2}L_{t-1}^{G,i}$$

The first inequality (1) holds due to $e^x+e^{-x}$ being monotone with respect to $|x|$, inequality (2) follows from the fact that for $0\leq|x|\leq\frac{1}{2}$, $\exp(x)\leq 1+x+2x^2$, and $\left|\eta\cdot\frac{v_\delta(q_t,s_t)}{f(n_{t-1}^{G,i})}\right|\leq\frac{1}{2}$ because of the way we set $\eta\in(0,1/2)$ and the fact that $|v_\delta(q_t,s_t)|\in[0,1]$ and $f(n_{t-1}^{G,i})\geq 1$.

Therefore, we have the desired bound

$$L_t-L_{t-1}=\sum_{(G,i)\in A_t(\pi_t)}L_t^{G,i}-L_{t-1}^{G,i}\leq\sum_{(G,i)\in A_t(\pi_t)}\eta v_\delta(q_{t+1},s_{t+1})C_{t-1}^{G,i}+\frac{2\eta^2}{f(n_t^{G,i})^2}L_t^{G,i}.\quad\square$$

## A.2  Proof of Lemma 3.2

*Proof.* For any threshold $q$, let $i_q$ be the bucket index such that $q\in B_m(i_q)$. For simplicity, define

$$u(q,s)=v_\delta(q,s)\sum_{(G,i)\in A_t(\pi_t)}C_t^{G,i}=v_\delta(q,s)C_t^q$$

where we overloaded the notation to write

$$C_t^q=C_t^{i_q}=\sum_{G\in\mathcal{G}(x_t)}C_t^{G,i_q}.$$

**Case (i)** $C_{t-1}^i<0$ **for all** $i\in[n]$:  With $q_t=1$, we have

$$\mathop{\mathbb{E}}_{q_t\sim Q^L,s_t\sim Q^A}[u(q_t,s_t)|x_t]=C_{t-1}^1(x_t)\mathop{\mathbb{E}}_{s_t\sim Q^A}[v_\delta(1,s_t)|x_t]<0$$

as $v_\delta(1,d_t)=1-(1-\delta)>0$.

**Case (ii)** $C_{t-1}^i>0$ **for all** $i\in[n]$:  With $q_t=0$, we have

$$\mathop{\mathbb{E}}_{q_t\sim Q^L,s_t\sim Q^A}[u(q_t,s_t)|x_t]=C_{t-1}^0(x_t)\mathop{\mathbb{E}}_{s_t\sim Q^A}[v_\delta(0,s_t)|x_t]<\rho C_{t-1}^0(x_t)<\rho L_{t-1}.$$

as we have

$$\mathop{\mathbb{E}}_{s_t\sim Q^A}[\text{Cover}(0,s_t)|x_t]-(1-\delta)\leq\mathop{\mathbb{E}}_{s_t\sim Q^A}[\text{Cover}(0,s_t)|x_t]=\mathop{\text{Pr}}_{s_t\sim Q^A}[s_t=0|x_t]\leq\rho$$

**Case (iii) there exists** $i^*\in[n-1]$ **such that** $C_{t-1}^{i^*}\cdot C_{t-1}^{i^*+1}\leq 0$:  First, consider the case where $C_{t-1}^{i^*}\geq 0$ and $C_{t-1}^{i^*+1}\leq 0$. Then, we have

$$\mathop{\mathbb{E}}_{q_t\sim Q^L,s_t\sim Q^A}[u(q_t,s_t)|x_t]$$

$$=p_t\mathop{\mathbb{E}}_{s_t\sim Q^A}\left[u\left(\frac{i^*}{n}-\frac{1}{rn},s_t\right)\middle|x_t\right]+(1-p_t)\mathop{\mathbb{E}}_{s_t\sim Q^A}\left[u\left(\frac{i^*}{n},s_t\right)\middle|x_t\right]$$

$$=p_tC_{t-1}^{i^*}(x_t)\mathop{\mathbb{E}}_{s_t\sim Q^A}\left[v_\delta\left(\frac{i^*}{n}-\frac{1}{rn},s_t\right)\middle|x_t\right]+(1-p_t)C_{t-1}^{i^*+1}\mathop{\mathbb{E}}_{s_t\sim Q^A}\left[v_\delta\left(\frac{i^*}{n},s_t\right)\middle|x_t\right]$$

$$\leq p_t C_{t-1}^{i^*} \left( \operatorname*{\mathbb{E}}_{s_t \sim Q^A} \left[ v_\delta \left( \frac{i^*}{n}, s_t \right) \Big| x_t \right] \right) + (1 - p_t) C_{t-1}^{i^*+1} \operatorname*{\mathbb{E}}_{s_t \sim Q^A} \left[ v_\delta \left( \frac{i^*}{n}, s_t \right) \Big| x_t \right]$$

$$= \operatorname*{\mathbb{E}}_{s_t \sim Q^A} \left[ v_\delta \left( \frac{i^*}{n}, s_t \right) \Big| x_t \right] \left( p_t C_{t-1}^{i^*} + (1 - p_t) C_{t-1}^{i^*+1} \right)$$

$$= 0.$$

The first inequality follows from the fact that $\operatorname{Cover}(\frac{i^*}{n} - \frac{1}{rn}, s) \leq \operatorname{Cover}(\frac{i^*}{n}, s)$ for any $s$. Now, consider the other case where $C_{t-1}^{i^*} \leq 0$ and $C_{t-1}^{i^*+1} \geq 0$.

$$\operatorname*{\mathbb{E}}_{q_t \sim Q^L, s_t \sim Q^A} [u(q_t, s_t)|x_t]$$

$$= p_t \operatorname*{\mathbb{E}}_{s_t \sim Q^A} \left[ u \left( \frac{i^*}{n} - \frac{1}{rn}, s_t \right) \Big| x_t \right] + (1 - p_t) \operatorname*{\mathbb{E}}_{s_t \sim Q^A} \left[ u \left( \frac{i^*}{n}, s_t \right) \Big| x_t \right]$$

$$= p_t C_{t-1}^{i^*}(x_t) \operatorname*{\mathbb{E}}_{s_t \sim Q^A} \left[ v_\delta \left( \frac{i^*}{n} - \frac{1}{rn}, s_t \right) \Big| x_t \right] + (1 - p_t) C_{t-1}^{i^*+1} \operatorname*{\mathbb{E}}_{s_t \sim Q^A} \left[ v_\delta \left( \frac{i^*}{n}, s_t \right) \Big| x_t \right]$$

$$\leq p_t C_{t-1}^{i^*}(x_t) \operatorname*{\mathbb{E}}_{s_t \sim Q^A} \left[ v_\delta \left( \frac{i^*}{n} - \frac{1}{rn}, s_t \right) \Big| x_t \right] + (1 - p_t) C_{t-1}^{i^*+1} \left( \operatorname*{\mathbb{E}}_{s_t \sim Q^A} \left[ v_\delta \left( \frac{i^*}{n} - \frac{1}{rn}, s_t \right) \Big| x_t \right] + \rho \right)$$

$$= \rho(1 - p_t) C_{t-1}^{i^*+1} + \operatorname*{\mathbb{E}}_{s_t \sim Q^A} \left[ v_\delta \left( \frac{i^*}{n} - \frac{1}{rn}, s_t \right) \Big| x_t \right] \left( p_t C_{t-1}^{i^*} + (1 - p_t) C_{t-1}^{i^*+1} \right)$$

$$= \rho L_{t-1}.$$

The first inequality follows from the fact that

$$\operatorname*{\operatorname{Pr}}_{s_t \sim Q^A} \left[ \operatorname{Cover} \left( \frac{i^*}{n}, s_t \right) \Big| x_t \right] - \operatorname*{\operatorname{Pr}}_{s_t \sim Q^A} \left[ \operatorname{Cover} \left( \frac{i^*}{n} - \frac{1}{rn}, s_t \right) \Big| x_t \right]$$

$$\leq \operatorname*{\operatorname{Pr}}_{s_t \sim Q^A} \left[ s_t \in \left[ \frac{i^*}{n} - \frac{1}{rn}, \frac{i^*}{n} \right] \Big| x_t \right]$$

$$\leq \rho.$$

$\square$

### A.3 Proof of Theorem 3.1

*Proof.* Fix any round $t \in [T]$ and transcript $\pi_{1:t-1}$. For simplicity, we write $L_t = L_t(\pi_{1:t})$. Then, we can use Lemma 3.1 to prove the following lemma.

**Lemma A.1.** *Fix any transcript $\pi_{1:T}$. Then, for any round $t \in [T]$, we have*

$$L_t \leq L_{t-1} \left( 1 + \frac{\eta v_\delta(q_t, (x_t, s_t))}{L_{t-1}} \sum_{(G,i) \in A_t(\pi_t)} C_{t-1}^{G,i} + \sum_{(G,i) \in A_t(\pi_t)} \frac{2\eta^2}{f(n_t^{G,i})^2} \right).$$

*Proof.* Fix transcript $\pi_{1:T}$. Then at any round $t$, we have

$$L_t$$
$$= L_{t-1} + L_t - L_{t-1}$$
$$\leq L_{t-1} + \sum_{(G,i) \in A_t(\pi_t)} \eta v_\delta(q_t, (x_t, s_t)) C_{t-1}^{G,i} + \frac{2\eta^2}{f(n_t^{G,i})^2} L_{t-1}^{G,i} \qquad \text{(Lemma 3.1)}$$
$$\leq L_{t-1} + \sum_{(G,i) \in A_t(\pi_t)} \eta v_\delta(q_t, (x_t, s_t)) C_{t-1}^{G,i} + L_{t-1} \sum_{(G,i) \in A_t(\pi_t)} \frac{2\eta^2}{f(n_t^{G,i})^2} \qquad \left( L_{t-1}^{G,i} \leq L_{t-1} \right)$$
$$\leq L_{t-1} \left( 1 + \frac{\eta v_\delta(q_t, (x_t, s_t))}{L_{t-1}} \sum_{(G,i) \in A_t(\pi_t)} C_{t-1}^{G,i} + \sum_{(G,i) \in A_t(\pi_t)} \frac{2\eta^2}{f(n_t^{G,i})^2} \right).$$

$\square$

Applying Lemma A.1 recursively, we get

$$L_T \leq L_0 \prod_{t=1}^{T} \left( 1 + \frac{\eta v_\delta(q_t, (x_t, s_t))}{L_{t-1}} \sum_{(G,i) \in A_t(\pi_t)} C_{t-1}^{G,i} + \sum_{(G,i) \in A_t(\pi_t)} \frac{2\eta^2}{f(n_t^{G,i})^2} \right)$$

$$\overset{(3)}{\leq} L_0 \prod_{t=1}^{T} \exp \left( \frac{\eta v_\delta(q_t, (x_t, s_t))}{L_{t-1}} \sum_{(G,i) \in A_t(\pi_t)} C_{t-1}^{G,i} + \sum_{(G,i) \in A_t(\pi_t)} \frac{2\eta^2}{f(n_t^{G,i})^2} \right)$$

$$\leq L_0 \exp \left( \sum_{t=1}^{T} \frac{\eta v_\delta(q_t, (x_t, s_t))}{L_{t-1}} \sum_{(G,i) \in A_t(\pi_t)} C_{t-1}^{G,i} + \sum_{t=1}^{T} \sum_{(G,i) \in A_t(\pi_t)} \frac{2\eta^2}{f(n_t^{G,i})^2} \right)$$

$$\leq L_0 \exp \left( \sum_{t=1}^{T} \frac{\eta v_\delta(q_t, (x_t, s_t))}{L_{t-1}} \sum_{(G,i) \in A_t(\pi_t)} C_{t-1}^{G,i} + \sum_{G \in \mathcal{G}, i \in [m]} \sum_{n=1}^{n_T^{G,i}} \frac{2\eta^2}{f(n)^2} \right)$$

$$\leq L_0 \exp \left( \sum_{t=1}^{T} \frac{\eta v_\delta(q_t, (x_t, s_t))}{L_{t-1}} \sum_{(G,i) \in A_t(\pi_t)} C_{t-1}^{G,i} + \sum_{G \in \mathcal{G}, i \in [m]} \sum_{n=1}^{\infty} \frac{2\eta^2}{f(n)^2} \right)$$

$$\leq L_0 \exp \left( \sum_{t=1}^{T} \frac{\eta v_\delta(q_t, (x_t, s_t))}{L_{t-1}} \sum_{(G,i) \in A_t(\pi_t)} C_{t-1}^{G,i} + 2\eta^2 K_\epsilon |\mathcal{G}| m \right)$$

$$= 2|\mathcal{G}| m \exp \left( \sum_{t=1}^{T} \frac{\eta v_\delta(q_t, (x_t, s_t))}{L_{t-1}} \sum_{(G,i) \in A_t(\pi_t)} C_{t-1}^{G,i} + 2\eta^2 K_\epsilon |\mathcal{G}| m \right)$$

where inequality (3) follows from $1 + x \leq \exp(x)$.

Taking the log of both sides, we have

$$\ln(L_T) \leq \ln(2|\mathcal{G}| m) + \sum_{t=1}^{T} \frac{\eta v_\delta(q_t, (x_t, s_t))}{L_{t-1}} \sum_{(G,i) \in A_t(\pi_t)} C_{t-1}^{G,i} + 2\eta^2 K_\epsilon |\mathcal{G}| m$$

for any $\pi_{1:T}$.

By Observation 3.1, it suffices to upper bound $\max_{G \in \mathcal{G}, i \in [m]} \frac{|V_T^{G,i}|}{f(n_T^{G,i})}$. We have:

$$\max_{G \in \mathcal{G}, i \in [m]} \frac{|V_T^{G,i}|}{f(n_T^{G,i})} = \frac{1}{\eta} \ln \left( \exp \left( \max_{G \in \mathcal{G}, i \in [m]} \frac{\eta |V_T^{G,i}|}{f(n_T^{G,i})} \right) \right)$$

$$= \frac{1}{\eta} \ln \left( \max_{G \in \mathcal{G}, i \in [m]} \exp \left( \frac{\eta |V_T^{G,i}|}{f(n_T^{G,i})} \right) \right)$$

$$\leq \frac{1}{\eta} \ln \left( \sum_{G \in \mathcal{G}, i \in [m]} \exp \left( \frac{\eta |V_T^{G,i}|}{f(n_T^{G,i})} \right) \right)$$

$$\leq \frac{1}{\eta} \ln \left( \sum_{G \in \mathcal{G}, i \in [m]} \exp \left( \frac{\eta V_T^{G,i}}{f(n_T^{G,i})} \right) + \exp \left( \frac{-\eta V_T^{G,i}}{f(n_T^{G,i})} \right) \right)$$

$$= \frac{\ln(L_T)}{\eta}$$

$$\leq \frac{1}{\eta} \left( \ln(2|\mathcal{G}| m) + \sum_{t=1}^{T} \frac{\eta v_\delta(q_t, (x_t, s_t))}{L_{t-1}} \sum_{(G,i) \in A_t(\pi_t)} C_{t-1}^{G,i} + 2\eta^2 K_\epsilon |\mathcal{G}| m \right).$$

Taking expectation over $\pi_{1:T}$ on both sides, we get

$$\mathop{\mathbb{E}}_{\pi_{1:T}}\left[\max_{G\in\mathcal{G},i\in[m]}\frac{|V_T^{G,i}|}{f(n_T^{G,i})}\right]$$

$$\leq \mathop{\mathbb{E}}_{\pi_{1:T}}\left[\frac{1}{\eta}\left(\ln(2|\mathcal{G}|m)+\sum_{t=1}^{T}\frac{\eta v_\delta(q_t,(x_t,s_t))}{L_{t-1}}\sum_{(G,i)\in A_t(\pi_t)}C_{t-1}^{G,i}+2\eta^2 K_\epsilon|\mathcal{G}|m\right)\right]$$

$$\leq \frac{1}{\eta}\left(\ln(2|\mathcal{G}|m)+2\eta^2 K_\epsilon|\mathcal{G}|m+\mathop{\mathbb{E}}_{\pi_{1:T}}\left[\sum_{t=1}^{T}\frac{\eta v_\delta(q_t,(x_t,s_t))}{L_{t-1}}\sum_{(G,i)\in A_t(\pi_t)}C_{t-1}^{G,i}\right]\right).$$

Let us focus only on the third term:

$$\mathop{\mathbb{E}}_{\pi_{1:T}}\left[\sum_{t=1}^{T}\frac{\eta v_\delta(q_t,(x_t,s_t))}{L_{t-1}}\sum_{(G,i)\in A_t(\pi_t)}C_{t-1}^{G,i}\right]$$

$$=\mathop{\mathbb{E}}_{\pi_{1:T-1}}\left[\mathop{\mathbb{E}}_{\pi_T}\left[\sum_{t=1}^{T}\frac{\eta v_\delta(q_t,(x_t,s_t))}{L_{t-1}}\sum_{(G,i)\in A_t(\pi_t)}C_{t-1}^{G,i}\Bigg|\pi_{1:T-1}\right]\right]$$

$$=\mathop{\mathbb{E}}_{\pi_{1:T-1}}\left[\sum_{t=1}^{T-1}\frac{\eta v_\delta(q_t,(x_t,s_t))}{L_{t-1}}\sum_{(G,i)\in A_t(\pi_t)}C_{t-1}^{G,i}+\frac{\eta}{L_{T-1}}\mathop{\mathbb{E}}_{\pi_T}\left[v_\delta(q_T,(x_T,s_T))\sum_{(G,i)\in A_T(\pi_T)}C_{T-1}^{G,i}\Bigg|\pi_{1:T-1}\right]\right]$$

$$\overset{(4)}{\leq}\mathop{\mathbb{E}}_{\pi_{1:T-1}}\left[\sum_{t=1}^{T-1}\frac{\eta v_\delta(q_t,(x_t,s_t))}{L_{t-1}}\sum_{(G,i)\in A_t(\pi_t)}C_{t-1}^{G,i}+\eta\rho\right]$$

$$\leq \dots$$
$$\leq \eta\rho T$$

where inequality (4) comes from Lemma 3.2.

In other words, we have

$$\mathop{\mathbb{E}}_{\pi_{1:T}}\left[\max_{G\in\mathcal{G},i\in[m]}\frac{|V_T^{G,i}|}{f(n_T^{G,i})}\right]\leq \frac{1}{\eta}\left(\ln(2|\mathcal{G}|m)+2\eta^2 K_\epsilon|\mathcal{G}|m+\eta\rho T\right)$$

$$=\frac{\ln(2|\mathcal{G}|m)}{\eta}+2\eta|\mathcal{G}|mK_\epsilon+\rho T$$

$$\leq \sqrt{4K_\epsilon|\mathcal{G}|m\ln(|\mathcal{G}|m)}+\rho T$$

where the last inequality follows from setting $\eta=\sqrt{\frac{\ln(|\mathcal{G}|m)}{2K_\epsilon|\mathcal{G}|m}}$. Note that $\eta < 1/2$ as $2\ln(|\mathcal{G}|m) < K_\epsilon|\mathcal{G}|m$ because $K_\epsilon \geq 1$.

$\square$

# B  Experiments

In this section, we evaluate MVP and compare it to more traditional methods of conformal prediction on a variety of tasks. In each comparison, we use the same model and conformal score for MVP and for the methods we compare against — the only difference is the type of the conformal prediction wrapper. Our code is available at https://github.com/ProgBelarus/MultiValidPrediction.

First in Section B.1 we study a synthetic regression problem in a simple exchangeable (i.i.d.) setting, and compare to split conformal prediction [Lei et al., 2018]. We show that even when we measure only marginal empirical coverage, MVP improves over split conformal prediction when the regression function must be learned. This is because to maintain the exchangeability of conformal scores, split conformal prediction must split the data into two sets — one for training the regression function and

one for calibrating the prediction sets.[B.1] In contrast, our method does not require exchangeability, so we can both train the regression model and calibrate our prediction sets on the entire dataset. Then, we modify our regression problem so that there are 20 overlapping sub-populations, and one of the sub-populations (consisting of half of the data points) has higher label noise. We measure group-wise coverage for MVP, for naive split conformal prediction that has no knowledge of the groups to be covered, and the method of Foygel Barber et al. [2020] which guarantees (conservative) group-wise coverage for intersecting groups. We find that MVP significantly improves on both methods. Finally we run all three of these methods on real data drawn i.i.d. from a U.S. Census dataset provided by the Folktables package [Ding et al., 2021], where we ask for group-wise coverage on groups defined by race and sex designations. Again, we find that MVP consistently obtains the closest to its target group-wise coverage while providing narrower prediction intervals.

Next, in Section B.2 we study a regression problem in the presence of covariate shift. First we replicate an experiment of Tibshirani et al. [2019], in which a synthetic covariate shift (with known propensity scores and known changepoint) is simulated on a UCI dataset. The method of Tibshirani et al. [2019] reweights the calibration set using the propensity scores. MVP can also take advantage of propensity scores when they are known: we give MVP a "warm start" from the same portion of the dataset that split conformal prediction uses for calibration, sampled with replacement after being re-weighted by the propensity scores. Both algorithms are then evaluated on the shifted distribution. We find both algorithms perform comparably. We then experiment with unknown and unanticipated covariate shift simulated on datasets derived by U.S. Census data provided from the Folktables package [Ding et al., 2021]: We compare to split conformal prediction calibrated on the California data (this time without re-weighting) and evaluated on the Pennsylvania data. Similarly, we again give MVP a warm start on the California data (again without reweighting), and then measure its performance on 2018 Pennsylvania Census data. We find that MVP obtains the correct coverage rate and smaller interval widths compared to the split conformal method despite having no knowledge of the distribution shift.

In Section B.3 we evaluate MVP on time series data — 20 years of stock returns, in a volatility prediction task. We compare MVP to the Adaptive Conformal Inference (ACI) method of Gibbs and Candes [2021], which guarantees marginal (but not threshold calibrated) coverage for adaptively chosen data. When evaluated in terms of marginal coverage, we find that MVP and ACI perform comparably: ACI obtains average coverage slightly closer to the target, whereas MVP predicts a more stable sequence of thresholds. We then complicate the experiment to exhibit the two advantages of MVP (groupwise coverage and threshold calibrated coverage). First we define 20 intersecting groups defined as the trading days that are multiples of $1, 2, \ldots, 20$. We add perturbations to the stock returns that differ across these groups, and find that MVP continues to produce the correct group-wise coverage, whereas ACI fails to. Next, we produce a fully adversarial sequence by presenting examples to the algorithms not in time order but in *sorted order by their conformal scores*. By construction, this sequence would cause split conformal prediction methods to have 0 coverage, but both ACI and MVP are required to obtain the correct marginal coverage on this sequence. However, we find that given this sequence, ACI reduces to a strategy that, similar to the uninformative "cheating" strategy mentioned in the Introduction, predicts the trivial coverage interval (all of $[0, 1]$) on most days — which guarantees marginal but not threshold calibrated coverage, and does not produce non-trivial average interval widths. In contrast, MVP, by virtue of its threshold calibration condition, produces a sequence of coverage thresholds that correctly track the sequence of conformal scores of the true labels in the data, and hence produces prediction intervals with the correct widths.

Finally in Section B.4 we compare MVP to the work of Angelopoulos et al. [2020] on a large-scale ImageNet classification task. We find that MVP obtains comparable coverage rates and prediction set sizes, despite the fact that the setting is favorable to Angelopoulos et al. [2020] — i.e. the data is i.i.d. and we measure only marginal coverage.

## B.1 Exchangeable Data

### B.1.1 Basic Experimental Setup and Marginal Coverage

We simulate a synthetic linear regression problem in which the regression model must be trained in tandem with the conformal predictor. The feature domain consists of 10 binary features and 290

---

[B.1]This is not just a theoretical requirement — split conformal prediction fails badly otherwise.

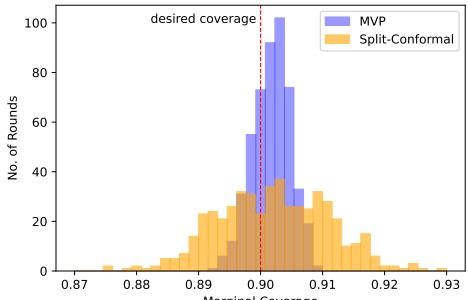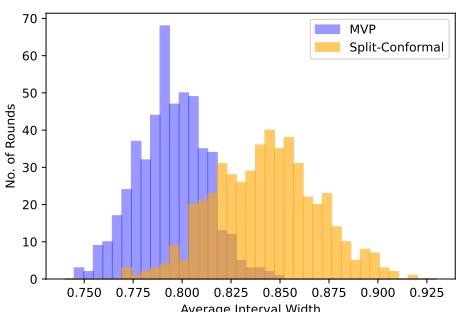

Figure B.1: The plot on the left is a histogram of the empirical marginal coverage of MVP and split conformal prediction over 500 repeated trials; the right hand plot is similarly a histogram of the average interval width for both methods. We see that MVP gets both empirical coverage that is more tightly concentrated around its target (0.9) and narrower coverage interval width.

continuous features. For any input $x$, the binary features are drawn from a uniform distribution and each continuous feature is drawn from a normal distribution $\mathcal{N}(0, \sigma_x^2)$. Each example's label is governed by an ordinary least squares model:

$$y = \langle \theta, x \rangle + \mathcal{N}(0, \sigma_y^2)$$

for some fixed vector $\theta \in \mathbb{R}^{300}$ unknown to the learner.

We run both MVP and split conformal prediction [Lei et al., 2018] using the conformal score $s_t(x, y) = |f_t(x) - y|$. When running MVP, we train $f_t$ using least squares regression on all points $(x_{t'}, y_{t'})$ for $t' < t$. For split conformal prediction, we divide points evenly between a calibration set and a training set (points from odd time steps go into the calibration set, points from even time steps go into the training set), and $f_t$ is trained using least squares regression on all points in the training set at time $t - 1$. (We also tried training $f_t$ on all points, but this causes split conformal prediction to fail catastrophically).

**Results** We set $\sigma_x^2 = 0.1$, $\sigma_y^2 = 0.2$ and run 500 independent trials of our experiment, each for $T = 2000$ steps. $\theta$ is independently selected for each trial. The results are shown in Figure B.1. MVP simultaneously obtains empirical coverage that is more tightly concentrated around its target and obtains narrower coverage intervals compared to split conformal prediction. Despite the fact that we are in a setting that is extremely favorable to split conformal prediction (i.i.d. data and marginal coverage evaluation), MVP has the advantage that it can use a regression function $f_t$ trained on *all* past data, without the need to set aside a calibration set. This is needed for split conformal prediction to maintain the exchangeability of the conformal scores.

### B.1.2 Multi-Group Coverage

We now compare the coverage of MVP to that of split conformal prediction not just marginally, but group-wise. We use the same feature generation process and conformal score as for our marginal coverage experiment described in Section B.1.1. Recall that the first 10 of the 300 features in our data domain are binary, which we now use to define 20 (intersecting) groups defined by the value of each of the 10 binary features. Labels are still generated according to an ordinary least squares model, but now the noise rate depends on the groups that each datapoint is a member of. Specifically:

$$y = \langle \theta, x \rangle + \mathcal{N}\left(0, \sigma^2 + \sum_{i=1}^{10} \sigma_i^2 x_i\right)$$

for some fixed vector $\theta \in \mathbb{R}^{300}$, and for fixed values of $\sigma_i$, each associated with one of the binary features indicating groups.

We run MVP parameterized to promise multi-valid coverage for the set of 20 intersecting groups defined by the first 10 binary features of the input: For each $i \in \{0, 1, \cdots, 19\}$, we define $G_i = \{x \in$

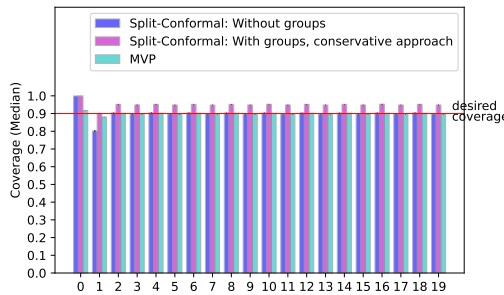
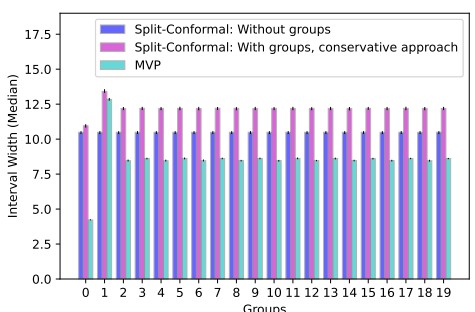

Figure B.2: On the left we plot the median over 100 independent trials of the coverage conditional on membership in each of our 20 groups. On the right we plot the median of the average interval width conditional on membership in each of the 20 groups. Compared to the split conformal prediction methods, we see that MVP obtains the target coverage level on each group (neither under nor over covering), and obtains narrower interval widths. The error bars represent 25th and 75th quantiles, and they are not easily visible in this figure as they are quite narrow: for conformal with groups, both bar endpoints are within $\pm 0.0039$ from the median, for conformal without groups, within $\pm 0.0054$ from the median, and for MVP, within $\pm 0.0021$ from the median.

$\mathcal{X} \mid x_{\lceil (i+1)/2 \rceil} \equiv_2 i\}$ and let $\mathcal{G} = \{G_i \mid 0 \le i \le 19\}$. At each time-step $t$, we train a regression model $f_t$ on all past data.

We compare to two benchmark conformal prediction methods. First, we compare to naive split conformal prediction (which ignores the group structure), just as in Section B.1.1. This method offers no guarantees about group-wise coverage. Second, we compare to the method of Foygel Barber et al. [2020] which separately computes a calibration threshold for each of the 20 groups marginally, and on each example $x_t$, uses the most conservative (i.e. largest) threshold associated with any of the groups for which $x_t$ is a member. This method guarantees coverage *at least* the target coverage level, but does not guarantee coverage approaching the target. Note there are $2^{10}$ different subsets of groups that each example might be a member of, and so the method of Romano et al. [2020a] which separately calibrates on *disjoint* groupings of the data cannot be run without having roughly 1000-fold more data. For both conformal prediction methods we equally split the data between a training set used for training the regression model $f_t$ and a calibration set used for calibrating the prediction intervals. We run MVP with $m = 40$ calibration buckets.

**Results** We run 100 independent trials of our experiment, each for $T = 20,000$ data points. Our results are plotted in Figure B.2. We set $\sigma_1^2 = 3.0$ and $\sigma_2^2 = \ldots = \sigma_{10}^2 = 0.1$ so that $G_0$ is a "low noise" group and $G_1$ is a "high noise" group. We keep the values of the $\sigma_i$ fixed across all trials, but each is run with an independently drawn $\theta$. As expected, we find that naive split conformal prediction fails to meet its coverage target, over-covering on the low noise group and under-covering on the high noise group, and uses a uniform interval width. In contrast, both MVP and the conservative method of Foygel Barber et al. [2020] use different average interval widths for different groups. The conservative method of Foygel Barber et al. [2020] always gets at least the target coverage, but significantly over-covers on every group except for the high noise group. In contrast MVP obtains the target coverage on every group. MVP also has lower average interval width on every group compared to Foygel Barber et al. [2020], and (correctly) produces significantly narrower intervals on the low noise group.

### B.1.3 Multi-Group Coverage with Folktables Data

We now evaluate the group-wise performance of MVP against the same two split conformal prediction methods on a real dataset derived from the 2018 Census American Community Survey Public Use Microdata provided by the Folktables package [Ding et al., 2021]. The dataset includes instances of people from all the states in the USA; for this experiment, we consider only those instances from the state of California. There are 195665 instances of this kind, and we subsample this data (0.1 for training, 0.1 for testing).

Our goal in this experiment is to generate prediction sets for a person's income. The Folktables dataset has nine different codes for race[B.2] and two codes for sex[B.3]. Note that the race and sex groups intersect. We define groups for five out of nine of the race groups (the remaining four have very little data) and groups for both sexes, for a total of seven groups. We run MVP with $m = 40$ buckets, and parametrized to promise multi-valid coverage for each of these seven groups, and compare against both conformal prediction methods introduced in Section B.1.2.

Using the training data, we train a linear regression model $f$ to predict income and use it to define the conformal score $s(x, y) = |f(x) - y|$ for all three methods. An initial calibration set of size 1000 (taken from the test data) is used for both split conformal methods, and is used as a "warm start" for MVP (i.e. this data is used to update variables used in the algorithm, but we do not record performance over these instances). The remaining test data is used to compare performance between methods. For the split conformal methods, the calibration set grows to include the previously observed examples from the test set as time goes on.

**Results** We run 100 independent trials of our experiment with random subsampling of training and test data from the Folktables dataset. The results are shown in Figure B.3. While MVP obtains the desired coverage across all groups, the naive split conformal prediction method under-covers on some groups and over-covers on others, and the method of Foygel Barber et al. [2020] significantly over-covers on some groups ($G_3$, $G_4$ and $G_6$). Additionally, MVP consistently predicts smaller-width prediction intervals in comparison to both other methods.

## B.2 Covariate Shift

### B.2.1 Known Covariate Shift with UCI Airfoil Data

We first study the setting of known covariate shift considered by Tibshirani et al. [2019] (which introduced the weighted split conformal prediction method that we use as our point of comparison) and replicate their design. Following Tibshirani et al. [2019], we use the airfoil dataset from the UCI Machine Learning Repository [Dua and Graff, 2017] which consists of data of NACA 0012 airfoils. The dataset contains $N = 1503$ total instances of $d = 5$ features (frequency in Hz, angle of attack in degrees, chord length in meters, free-stream velocity in meters per second, and suction side displacement thickness in meters) The target feature for prediction is scaled sound pressure in decibels. In this setting, the data available for calibration is drawn from a different distribution from the data that is used for evaluation, but the distributions differ only in their relative weighting of feature vectors, and the relative weightings (likelihood ratios) are known.

Weighted split conformal prediction uses the likelihood ratios between the training and evaluation distributions to find weighted quantiles of the conformal scores on the evaluation data distribution. We note that MVP can also make use of these likelihood ratios when they are known. We do so by "warm starting" MVP by running it on the data that weighted split conformal prediction uses for calibration, but re-sampled with replacement using rejection sampling according to the known likelihood ratios[B.4].

Following the protocol in Tibshirani et al. [2019], for both methods, we use 25% of the data to train the underlying linear regression model that will be given to both MVP and weighted split conformal prediction. (It is necessary to use a separate split of the data for the method of Tibshirani et al. [2019], but for our method we could have shared data between training and calibration, which would give us an advantage of the sort we demonstrated in Section B.1. We do not do this in this experiment to disentangle different aspects of the comparison between our techniques). The weighted split conformal prediction algorithm is then given a calibration dataset of 25% of the data to compute the residual quantiles and finally samples with replacement 50% of the remaining points for the evaluation dataset, with probabilities proportional to $w(x) = \exp(x^T \beta)$, where $\beta = (-1, 0, 0, 0, 1)$. This final

---

[B.2]1. White alone, 2. Black or African American alone, 3. American Indian alone, 4. Alaska Native alone, 5. American Indian and Alaska Native tribes specified; or American Indian or Alaska Native, not specified and no other races, 6. Asian alone, 7. Native Hawaiian and other Pacific Islander alone, 8. Some Other Race alone, 9. Two or More Races.

[B.3]1. Male, 2. Female.

[B.4]We could have similarly reweighted the data in our potential function using the likelihood ratios, but we choose this method instead so as to apply our algorithm as a black box.

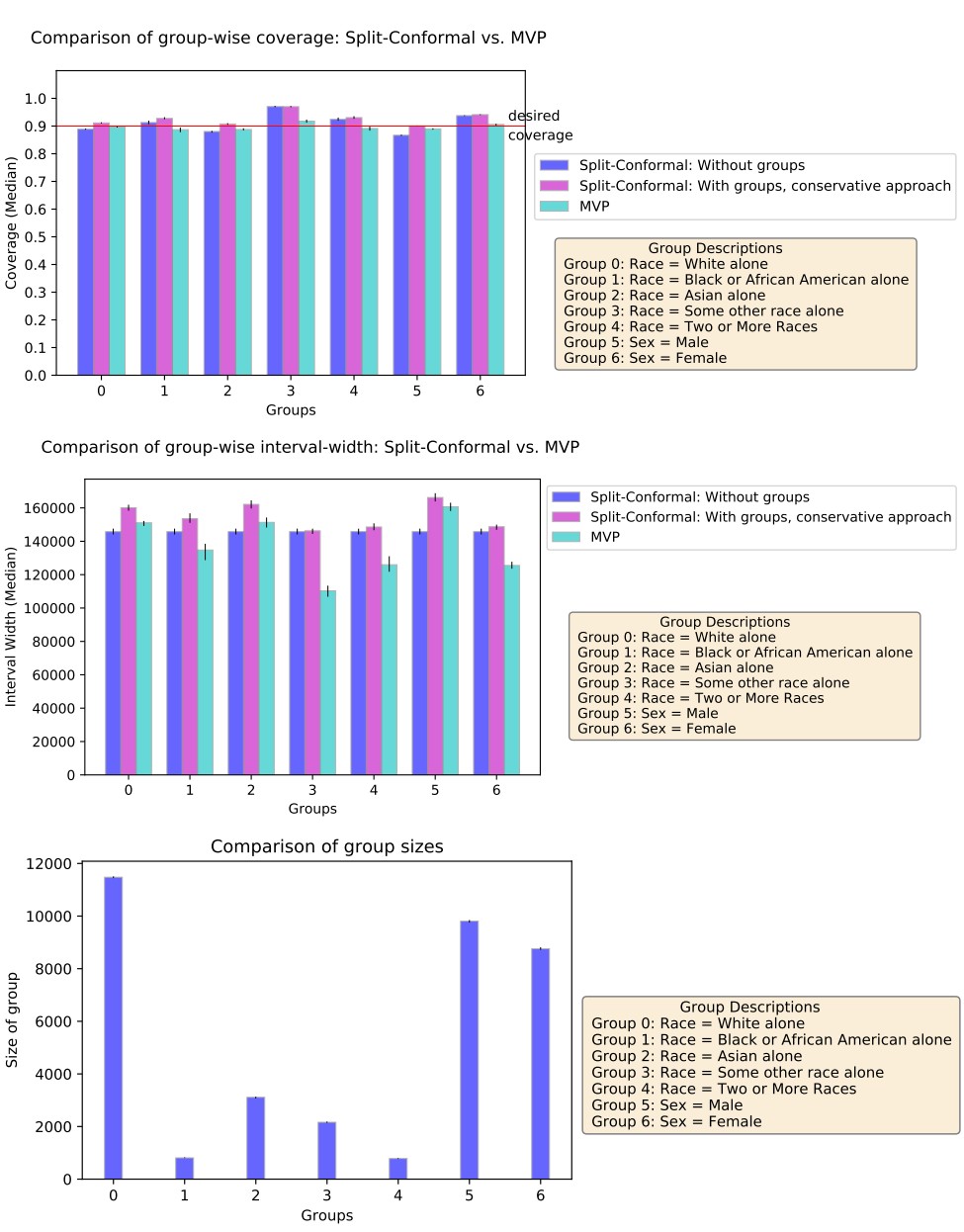

Figure B.3: The first plot shows the median over 100 independent trials of the marginal coverage conditional on membership in each group. The second plot shows the median of average interval width conditional on membership in each group. The third plot shows the average group size (number of elements in each group) over all 100 trials. Details about groups are to the right of each plot. The error bars represent 25th and 75th quantiles, and they are not easily visible in the first and third plot as they are quite narrow.

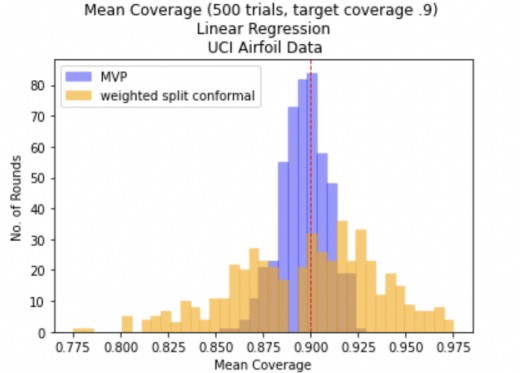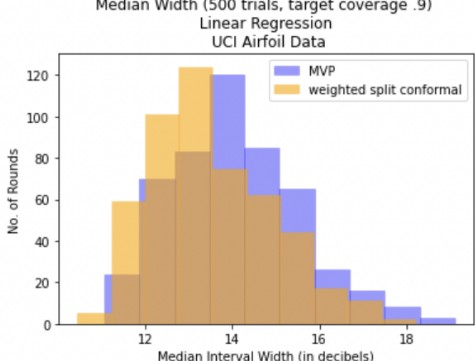

Figure B.4: The left-hand figure shows a histogram of the coverage rate of MVP and weighted split conformal prediction over 500 trials; the right-hand figure is a histogram of the median prediction interval widths over the same 500 trials.

fold simulates a synthetic covariate shift and in our comparison the weighted split conformal method that has oracle access to the shift likelihood ratios. When running MVP, we use this 25% of the dataset in a comparable way: we sample the calibration fold of the remaining data with replacement with probilities proportional to $w(x)$ and use it to run MVP as a warm start (i.e. the predictions that MVP makes on this fold are not recorded in the metrics we report). This uses the known conformal scores in a similar way to how they are used in weighted split conformal prediction. MVP is then evaluated on an evaluation dataset obtained the same way as for weighted split conformal, by sampling 50% of the remaining data with probabilities proportional to $w(x)$. We run MVP with $m = 40$ threshold-calibration buckets.

Figure B.4 shows a histogram of the coverage rate and median prediction interval width of both methods over 500 trials of the experiment, where each trial indicates a different train-test split of the data and a different sampling of the shifted data for the algorithm. We see that MVP obtains coverage that is significantly more tightly concentrated around its target (0.9) compared to weighted split conformal prediction and comparable interval widths. We note that this is even without letting MVP train its regression model on the calibration dataset.

### B.2.2 Unknown Shift with Folktables Data

Next we evaluate split conformal prediction and MVP on real data exhibiting distribution shift, in which the distribution changepoint and propensity scores are unknown (and so cannot be used to weight the calibration set as in our earlier experiment). Here we use the Folktables package [Ding et al., 2021]. The dataset consists of $N = 263973$ (CA: 195665, PA: 68308) instances each comprising of $d = 9$ features. We subsample the dataset (.4 of CA, .2 of PA), thus using $N = 91927$ overall. The features of the data are Census demographic attributes and the target prediction variable is income. We follow Ding et al. [2021] in investigating covariate shift that results from using data derived from different states related to the same task.

As in all of our experiments, MVP is trained using $m = 40$ buckets for calibration. For both MVP and split conformal prediction we use the quantile-regression based conformal score from Romano et al. [2019], using a quantile regression model trained on half of the available California data. We then use the remaining California data as the calibration dataset, used to "warm start" MVP and compute the residual quantiles for split conformal prediction. Finally, we evaluate MVP and split conformal on the Pennsylvania data and report a histogram of the empirical coverage and interval widths for both methods over 50 trials in Figure B.5. MVP comes very close to its coverage target (0.9), whereas split conformal prediction significantly over-covers. Similarly, MVP obtains narrower average prediction interval widths. Here the empirical coverage for both methods is much more tightly concentrated than it is for the UCI Airfoil dataset: this is because the dataset we are using in this experiment is roughly 60 times larger.

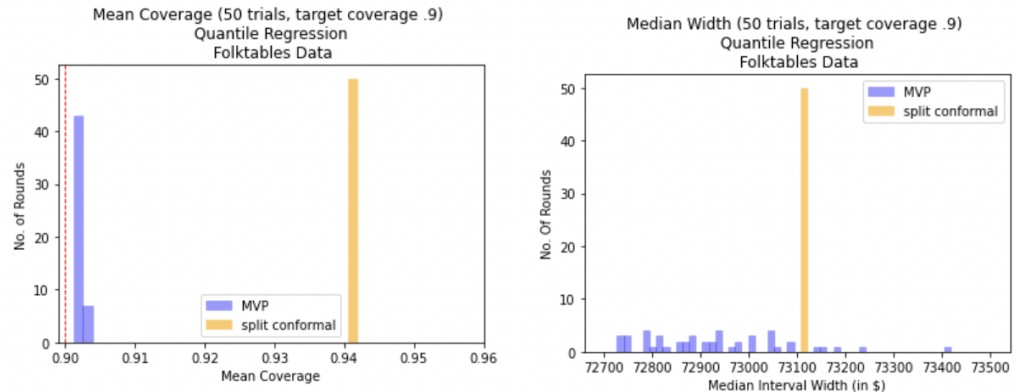

Figure B.5: The left-hand figure shows a histogram of the coverage for MVP and split conformal prediction over 50 trials; the right-hand figure shows a histogram of the prediction interval width.

## B.3 Time Series Data

### B.3.1 Basic Experimental Setup and Marginal Coverage

In this set of experiments, we run MVP on stock market data and compare our performance to the Adaptive Conformal Inference (ACI) algorithm of Gibbs and Candes [2021], a recent method that guarantees *marginal* coverage for adversarially chosen data. In contrast to MVP, ACI promises only marginal coverage (in particular, its guarantees are not threshold calibrated), and so we expect its convergence to be faster but that its thresholds will fluctuate more; our experiments bear this out.

To directly compare to ACI, we use the same dataset and model construction as in Gibbs and Candes [2021]. Specifically, we start with WSJ daily open price data[B.5] for AMD stock in 2000-2020 (corresponding to $T = 5283$ price points $p_1, \ldots, p_T$). We calculate daily returns $r_t$ as $r_t = \frac{p_t - p_{t-1}}{p_{t-1}}$ for every day $t$. Based on the returns, we then calculate the (realized) daily volatility as $v_t = r_t^2$ for $t \in [T]$. For our prediction task we train a model to estimate daily volatility levels $v_t$. Following Gibbs and Candes [2021], we use a standard sequential prediction model called GARCH [Bollerslev, 1986]; every day, GARCH makes volatility prediction $\sigma_t$, and autoregressively updates the model once it sees the realized volatility $v_t$. The conformal score we use on day $t$ is the *normalized regression score* $s_t(t, v) = |v_t - \sigma_t| / \sigma_t$. Here $\sigma_t$ is the prediction that the GARCH model makes at round $t$, and possible realizations of the volatility $v_t$ play the role of the label. We run MVP and ACI, for miscoverage target $\delta = 0.1$, on the (rescaled) scores[B.6] $\tilde{s}_1, \ldots, \tilde{s}_T$ of the GARCH model trained to predict AMD stock volatility. In all our experiments with ACI, we set the ACI hyperparameters as follows: $\gamma = 0.005$ (step size), lookback = 100, offset = 10. Figure B.6 shows the sequences of conformity thresholds for MVP and ACI. In general we find that even when we only measure marginal prediction, MVP performs comparably to ACI. Both methods obtain coverage close to the target rate of $0.9$, where ACI consistently gets a bit closer to the target rate. We visually observe that MVP makes more stable predictions compared to ACI, locally converging to a small stable set of threshold values (and moving over to the next stable set of thresholds once the scores have drifted sufficiently far), whereas ACI uses continuously fluctuating threshold values (this is expected, since it is not aiming for threshold calibrated coverage).

### B.3.2 Multigroup Coverage

Next, we augment the experimental setup to investigate multigroup coverage. We define a set of groups based on whether the index of the trading day is divisible by $1, \ldots, 20$: Define $x_t = t$, and let

---

[B.5] Available at https://www.wsj.com/market-data

[B.6] MVP assumes that the input scores $s_t \in [0, 1]$, but in this set of experiments we can only guarantee that the normalized regression score $s_t = |v_t - \sigma_t| / \sigma_t \in [0, \infty)$. Due to this, we feed MVP (and ACI, for consistency) modified scores $\tilde{s}_t = \frac{s_t}{1 + s_t} \in [0, 1]$. This type of rescaling works more generally in any setting where MVP's input scores belong to $[0, \infty)$ and need to be rescaled to be in $[0, 1]$: indeed, observe that the mapping $x \rightarrowtail \frac{x}{1+x}$ is a monotonic continuous bijection from $[0, \infty)$ to $[0, 1)$.

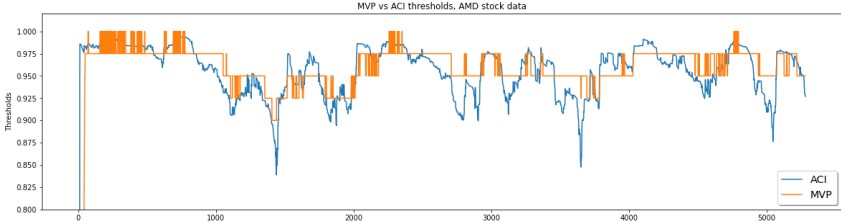

Figure B.6: A single trajectory of ACI and MVP thresholds plotted together; for convenience, threshold values are only displayed once MVP and ACI thresholds have risen above 0.8. One can see that MVP and ACI trajectories have somewhat similar shapes, but MVP exhibits a more stable behavior.

$\mathcal{G} = \{G_1, \ldots, G_{20}\}$, where $G_i$ is defined as the set of all $t$ such that $t \equiv 0 \mod i$. In other words, $G_1$ consists of the set of all time steps, $G_2$ consists of even time steps, $G_3$ consists of time-steps that are multiples of 3, and so on. As these sub-groups mutually intersect, it is not possible to run a separate copy of ACI on each one.

To provide sub-group variability, we artificially introduce varying levels of group-specific additive noise into the stock return data: for each $i \in [20]$, we add noise sampled from $\mathcal{N}(0, \hat{\sigma}_{\text{ret}})$ to the stock return $r_t$ on all days $t$ that fall into group $G_i$, where $\hat{\sigma}_{\text{ret}}$ is the empirical standard deviation of the returns sequence. This noise is additive: so the returns on a day that falls into multiple groups are perturbed by the sum of the group-specific perturbations.

We now run ACI and MVP on the scores produced by GARCH when it is trained on this noisy data. MVP is given the set of groups $\mathcal{G}$. Figure B.7 shows a plot of the median coverage rates (over 20 independent trials) for both ACI and MVP on each of the 20 groups. As expected, MVP achieves close to its target coverage on each group, whereas ACI — although getting very close to its target marginal coverage (see group 1) undercovers on most other groups, sometimes significantly as a result of the extra added noise.

### B.3.3 Adversarial Ordering

Finally, we present an experiment which tests MVP and ACI on a fully adversarial sequence of conformal scores: a sequence that linearly grows from 0.0 to 0.5 in $T = 5283$ equal steps[B.7], as shown in Figure B.8a. In this ordering, the next score is always larger than the algorithm has ever seen before — hence traditional conformal prediction methods that rely on the exchangeability assumption would obtain 0 coverage on this sequence.

As expected, ACI struggles when it sees scores that are always increasing. ACI and MVP are both guaranteed to approach the target marginal coverage, but this sequence serves to elucidate the difference between simple marginal coverage and threshold calibrated coverage. The trajectory of ACI's predicted thresholds on all rounds is shown in Figure B.8c. ACI's threshold oscillates rapidly between just below the current score and the maximum value (1) that its trajectory appears to fill the space between the score sequence and 1. We also show the histogram of ACI's thresholds over its full trajectory, which shows that most of them in fact correspond to the trivial prediction interval corresponding to the maximum threshold value. This reveals that on an increasing sequence, ACI obtains its target coverage by using a strategy that is very similar to the uninformative "cheating" strategy that we have discussed before: namely, ACI predicts the trivial prediction interval (all of $[0, 1]$) on many of the rounds, and periodically tries to predict lower threshold values (on which it miscovers and is forced back into predicting the full interval). These prediction intervals are not threshold calibrated. In contrast, MVP's sequence of predicted thresholds have to be threshold-calibrated hence (as shown in Figure B.8b) they closely track the actual score sequence, resulting in much more informative coverage intervals.

Beyond recognizing that ACI's thresholds, as opposed to MVP's, are uninformative in this setting, we can also see a concrete drawback of ACI's strategy by looking at its average prediction set width. Namely, suppose that the linearly increasing sequence of scores represent *regression scores*

---

[B.7]That is, the sequence is $\left\{ \frac{0.5 \cdot i}{T-1} \right\}_{t=0}^{T-1}$, where $T = 5283$ is chosen to be the same as in the above experiments

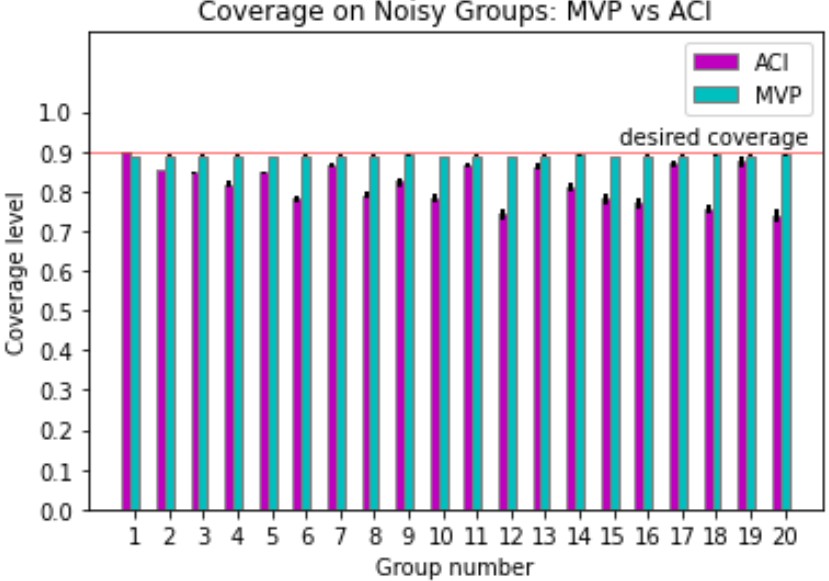

Figure B.7: MVP and ACI median coverage (over 20 indep. trials) on groups 1-20 on noisy data (group $j$ consists of days $t$ such that $t \equiv 0 \mod j$). MVP closely matches desired coverage level on all groups, whereas ACI significantly undercovers (within 10-20% from the target). In interpreting the plot, note that Group 1 consists of all of the rounds (and so represents overall marginal coverage), and that each group $j$ consists of a $1/j$ fraction of the data, so the groups become increasingly small from left to right. Note the very small (barely visible) error bars (spanning 25th to 75th quantile coverage): For ACI, the largest error quantile width across groups is $0.0303$, whereas for MVP it is even smaller: $0.007$.

$s_t(y_t, \hat{y}_t) = |y_t - \hat{y}_t|$ in a simple regression problem. Then, each threshold $q_t$ generated by ACI or MVP will produce an interval of width $2q_t$. In this case, the average width attained by MVP will be 0.526, whereas the average width attained by ACI will be 1.839. What is more, note that MVP's thresholds closely track the magnitude of the presented sequence of scores, while ACI's threshold is 1 most of the time no matter what subrange of $[0, 1]$ the observed scores are in. Therefore, if we generate increasing scores from 0 to $b$ (above, we took $b = 0.5$), where $b$ can be set arbitrarily small, we will get examples of adversarial data on which the prediction interval widths of ACI are *arbitrarily worse* than the prediction widths produced by MVP.

### B.4 A Classification Task: ImageNet

In this section, we compare the performance of MVP against an existing conformal prediction method for the task of generating prediction sets in image classification. The recent work of Angelopoulos et al. [2020] details and implements an algorithm, *Regularized Adaptive Prediction Sets* (RAPS) which, given a trained image classifier, generates small-sized prediction sets of image labels with marginal coverage guarantees. This is done by defining a modified conformal score which empirically produces smaller and more stable sets compared to previously used scores [Romano et al., 2020b].

Using ResNet-152 as the base image classifier, we use calibration data of size 1000 from ImageNet to train RAPS. For a fair comparison, we allow RAPS to be randomized like MVP by setting flag "randomized=True". We also have RAPS and MVP use the same hyperparameters for the conformal score. This same data is used as a "warm-start" training set for MVP (i.e. MVP predicts sets for this data and uses it to update variables used in the algorithm; MVP's performance over these time-steps is not recorded). MVP is run with $m = 40$ calibration buckets. The results shown in Figure B.9 detail the performance of both methods (using the same conformal score) on a held-out validation dataset of size 30,000.

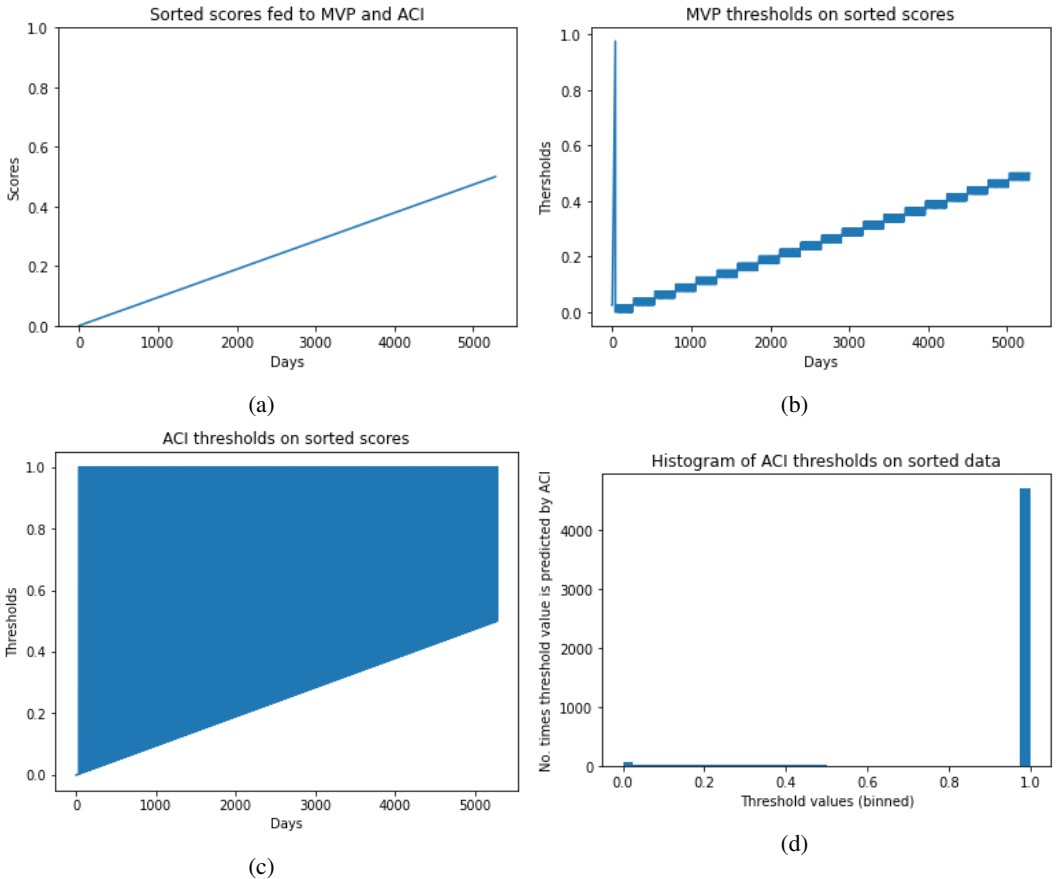

(a)

(b)

(c)

(d)

Figure B.8: MVP and ACI behavior on a sequence of sorted scores. Figure (a) plots the sequence of scores fed to both MVP and ACI. Figure (b) plots the sequence of thresholds chosen by MVP — note that it closely tracks the sequence of scores. Figure (c) plots the sequence of thresholds chosen by ACI. It appears to fill the upper diagonal region because it fluctuates so rapidly between the maximum value (1) and just below the score sequence. Figure (d) gives a histogram for the thresholds chosen by ACI, showing that ACI is almost always choosing the uninformative maximum threshold.

**Results** The marginal coverage achieved by RAPS across all $T = 30000$ images is 0.90523, and the marginal coverage achieved by MVP is 0.902133. The average prediction-set size for RAPS and MVP are 2.0506 and 2.13986 respectively, and the distribution across prediction-set sizes is similar for both methods. Once again we achieve competitive performance with "traditional" state of the art conformal prediction methods, even in a setting favorable to them (i.e. a setting with i.i.d. data in which only marginal coverage is measured).

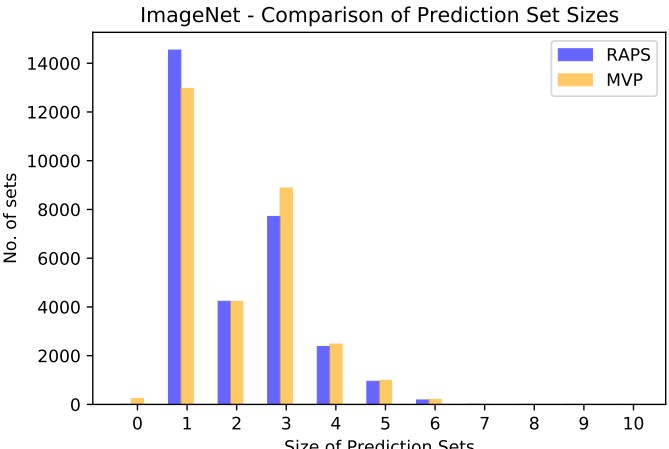

Figure B.9: A bar graph showing the size of prediction-sets generated by MVP and RAPS over a dataset of 30,000 images. MVP achieves prediction-set sizes on par with RAPS.