# OpenReview forum: "Practical Adversarial Multivalid Conformal Prediction"
_NeurIPS.cc/2022/Conference — NeurIPS 2022 Accept_

### Official Review · Reviewer_w6qw · 2022-07-04

**Rating:** 7
**Confidence:** 3
**Soundness:** 4 excellent
**Presentation:** 2 fair
**Contribution:** 3 good

**Summary:**

The paper presents a novel, lightweight conformal-inspired algorithm for quantifying uncertainty in an online setting. This is a good contribution: it is highly competitive and even outperforms existing methods for making sequential predictions with valid coverage, even under adversarial settings.

**Questions:**

See the comments under the weaknesses section.

**Limitations:**

The authors included a discussion about ethical considerations.

**Strengths And Weaknesses:**

**Strengths**

The proposed framework has several key advantages:

(1) It uses the entire data for training a predictive model and does not require using holdout data for calibration.

(2) The coverage guarantee holds in an adversarial setting.

(3) The coverage guarantee holds conditionally on the threshold used to form the prediction set.

(4) The coverage guarantee holds conditionally on possibly intersecting groups, which needed to be defined apriori.

(5) Numerical experiments demonstrate the validity of the theory, showing the advantage of this method in a wide variety of settings. In particular, I find the results presented in Section B.3 illuminating.

**Weaknesses**

(1)	The paper is written to experts in conformal prediction, and I am worried that this will hurt the deployment of the proposed method. Specifically,

a. Consider the introduction for example: what is a non-conformity score? What is $f$? What is $S_t$?

b. Next, moving to Algorithm 1: how the user should interpret each of the steps? Perhaps the authors can name, or explain in words the purpose of each step so that the reader will not lose intuition.

c. Consider Theorem 3.1: it asks to set $\eta $, but where $\eta $ is being used? It took me some time to understand that $\eta$ appears in the surrogate loss. This makes me wonder how does it affect the performance? How should the user set this parameter?

d. Consider pausing after Theorem 1 and explain it in words to ease the interpretation of this result.

e. Section 3.1 starts with a discussion about the differences between the proposed method and the one of Gupta et al. (2022). But, at this point, the reader is still trying to understand the implications of Theorem 1. Consider moving this discussion after sketching the proof.

(2) It is not clear how to check (online, as data arrive) that Definition 3.1 holds for a given triplet (\rho,r,m). I understand the statement that “We can also algorithmically enforce smoothness by perturbing the conformal scores with small amounts of noise from any continuous distribution, and so we should think of smoothness as a mild assumption”. But this will then reduce statistical efficiency, as algorithmic noise will be added to the scores. And also, how can we even know that we violate Definition 3.1?

(3) Section B.1: If I understand the experiment correctly, I believe the authors should also compare their method to ACI. Compared to split conformal, the gain in performance is due to the fact that MVP is applied online and re-fits the predictive model once a new test point is observed; this setting is more similar to the way that ACI (or its time-series version [ACI-TS]) is implemented.

[ACI-TS] Zaffran, Margaux, Aymeric Dieuleveut, Olivier Féron, Yannig Goude, and Julie Josse. "Adaptive conformal predictions for time series." arXiv preprint arXiv:2202.07282 (2022).

(4) Section B.2.1: same comment as above.

(5) Section B.4: same comment as above.

---

> ### Author Response · Authors · 2022-08-02
> **Response to Reviewer w6qw**
>
> Thank you for your careful review! Your feedback about the exposition resonates, and we will take your comments seriously as we revise the paper to improve clarity.
>
> **Choosing $\eta$:** In all of our experiments, we set $\eta$ as specified by Theorem 3.1, and so we do not treat it as a hyper-parameter in our work. It is true that it might be possible to further improve the experimental results by doing hyperparameter optimization to set $\eta$, but we do not explore this since the algorithm works well with the theoretically motivated setting for $\eta$, and this simplifies the algorithm by eliminating a hyperparameter.
>
> **Checking Definition 3.1:** It is true that we cannot verify from samples that a sequence of non-conformity scores is generated from a process that satisfies Definition 3.1. But, we can verify that MVP's coverage guarantees on a stream of data --- in all of our experiments, it obtains its target coverage guarantees without the need to separately perturb the nonconformity scores. Note that every continuous distribution with bounded density is $(\rho,r\cdot m)$ smooth for sufficiently large $r$, and our algorithm can be instantiated with an arbitrarily large value of $r$ without any cost in either run time or convergence; thus, we can always enforce the smoothness condition if needed by adding noise that is uniform in the interval $[0,\epsilon]$ for arbitrarily small $\epsilon$. If we do this, we continue to guarantee valid coverage of the original non-conformity scores by increasing the threshold at prediction time by the same (arbitrarily small) $\epsilon$. We will add a discussion to our paper to clarify this point, and note that we never need to do so in any of our experiments.
>
> **Comparisons to Split Conformal Prediction:** In our experiments, we use split conformal methods in an ``online mode'' --- as we make predictions in sequence, we add points that we have seen either to the calibration or training set, and retrain and re-calibrate the models at each step (except for the Imagenet experiment, in which we use a pre-trained model). So while you are right that it would be interesting to add ACI as a comparison, we are giving the split conformal methods the opportunity to make use of data as it arrives, just as we give to MVP.

---

> > ### Comment · Reviewer_w6qw · 2022-08-06
> > **follow-up**
> >
> > Thank you for the clarifications. While most of my concerns are resolved, it will be valuable to include experiments that demonstrate the effect of $\eta$ as well as robustness to violations of Definition 3.1. I recommend acceptance.

---

> > > ### Author Response · Authors · 2022-08-06
> > > **Thanks**
> > >
> > > Thank you!

---

### Official Review · Reviewer_RP3U · 2022-07-09

**Rating:** 7
**Confidence:** 4
**Soundness:** 3 good
**Presentation:** 2 fair
**Contribution:** 3 good

**Summary:**

The authors propose a calibration method in the spirit of (adaptive) conformal prediction for providing (conditional) coverage guarantees when prediction confidence intervals/sets without necessarily requiring exchangeable examples. Thus, the method is applicable to dependent time series data in addition to standard i.i.d. settings. In comparison to prior work, the method provide coverage conditional on arbitrary subgroups of inputs and conditional on the calibrated threshold in timer series data. To this end, an adversarial setting is assumed where the adversary chooses inputs and conformity scores from a distribution that is assumes to be reasonably smooth (i.e., not concentrated on individual points) – this is the only assumption made as far as I can tell.

**Questions:**

See weaknesses and conclusion above.

**Limitations:**

See weaknesses and conclusion above.

**Strengths And Weaknesses:**

Strengths:
- The paper is generally well-structured and notation is introduced clearly – even though the paper is very notation heavy in general, see weaknesses.
- The advantages of the proposed method are described fairly clearly in the intro.
- Generally, the paper addresses an important problem of (conditional) coverage guarantees without exchangeable data. While various papers have addressed individual parts (e.g., group-conditional coverage or calibration set conditional coverage), to the best of my knowledge, obtaining coverage conditioned on arbitrary groups as well as thresholds without requiring exchangeability is not possible so far.
- The authors demonstrate their approach in various settings similar to previous papers to demonstrate the method on various tasks involving standard classification and regression, time series data or covariate shift, mostly focusing on coverage and inefficiency (set/interval size).
- A sketch of the proof is provided in the paper.
- The appendix obtains sufficient details on experimental setup.

Weaknesses:
- The paper is very dense and notation-heavy. I appreciate that this is necessary for communicating the guarantee and a sketch of the proof, but I believe the authors could improve structure and writing that would make following the paper easier. Here are som individual points that I stumbled over:

    - Lines 39ff, the point of threshold-conditional coverage is not very clear imo.

    - Lines 56ff, this argument is also not very clear when the reader is unfamiliar with Gupta et al. - I skimmed through Gupta et al., but it is a very dense paper itself.

    - Footnote 1 seems rather important in terms of supporting one of the contributions – generally, I think a footnote is not the right place for this even if it might save space.

    - The introduction to notation in section 2 is helpful, but generally I would prefer it the authors would repeat the notation inline at appropriate places. Currently, nearly in every statement, I need to scroll back and check notation. Alternatively, a separate notation section could help, because sometimes it is also not clear where to scroll to to find the notation being introduced.

    - An example of the above is the definition of G_T(i) in definition 2.1.

    - Another example in K_\epsilon in the theorem.

    - Definition 2.1 is a bit unclear regarding the choice of \alpha – the choices provided seem somewhat arbitrary in the beginning and there is no discussion of this. It only becomes slightly clearer later in the paper.

    - Surrounding Definition 3.1, I think the authors could do a better job providing an intuition of the adversary. In the beginning it is very unclear what the adversary can do and what not. Also it is unclear what the adversary is supposed to model in practice (i.e., in practice, the scores are computed by the model and the adversary is just a way to model this adversarially in order to obtain a guarantee).

    - Algorithm 1 is typset too early in the paper in my opinion. How it works only becomes clear throughout the proof sketch. In fact, the proof sketch is used to introduce the method very indirectly. I spent half an our on the algorithm assuming that I would have to understand how it works, realizing later that the proof sketch is basically meant to walk me through it. I think this a big flaw structure and notation wise.

    - There should be more discussion on the theorem, what this bound means in practice, how different variables influence it – this is partly done in the following remark but in my opinion not sufficient to give a good understanding. Might also be because the analysis comes later.

    - 3.1 starts using notation from the algorithm, which was very hard to follow without reading 3.1 first. So this is a bit of a circle which should be broken in 3.1 or by moving the algorithm. It is also not made clear in the beginning that 3.1 will actually explains the algorithm.

    - Observation 3.1 is not straight forward. I feel this is by construction but should be worth 1-2 sentences of why this is.

    - In definition 3.4, the index t in L_t, V_t and \pi_t is unclear. Is the t used in V_t the same as in L_t or \pi_t (the arguments). This seems relevant as these seem disentangled in Lemma 3.1.

- Regarding related work, this paper seems to build on the work by Gupta et al. Generally, I would appreciate making the similarities clearer without requiring to read Gupta et al. But as the paper is already dense I would actually prefer not going into detail in comparison to Gupta et al. in the main paper but rather deferring that to the appendix and making this clear in the beginning of the main paper, this could also save some space. Nevertheless, can the authors give a concise discussion of the differences to Gupta et al. And what they built on/develop further? In the paper this is very spread and I was unable to get a clear picture.
- I think that split conformal prediction does provide calibration-set conditional coverage as discussed in detail in [a] and [b] and I think discussing these results would be important. This is because, assuming exchangeability (I know that this work goes beyond exchangeability), the benefit of MVP over split conformal prediction is limited – both are able to obtain calibration set and group-conditional coverage (when groups are known which AFAIK is the case here, too). Can the authors comment on this? (I acknowledge that [b] is very recent and more like concurrent work)

[a] Vovk. Conditional Validity of Inductive Conformal Predictors
[b] Bian et al. Training-conditional coverage for distribution-free predictive inference

- In figure 2, the groups seem to be very easy, split conformal prediction also provides conditional coverage except for one group. For me it looks like the groups are not really interesting – usually I would expect groups to obtain widely varying coverage levels if these groups are aligned somewhat with difficulty (like classes, fairness attributes or so).
- In the time series experiments, I am not sure whether I understand the motivation of adding noise. Can the authors comment on this? Is this just to make the task harder and show that MVP works better (because it doesn’t work better without noise)?
- In figure 3, what are the average lengths for both methods? Also, split conformal prediction always predicting the same interval is mainly due to the coarse histogram, right?
- In figure 4 a, shouldn’t there be any groups where ACI is overcovering. Marginal coverage is guaranteed but it underestimates coverage on all of them. Am I missing something?
- On ImageNet, the calibration set is very small, and RAPS still performs better (and RAPS is even less efficient then simpler conformity scores, see [c]). Does this mean that MVP is generally less relevant in split settings with exchangeability? Or does MVP still obtain better class/group-conditional coverage on ImageNet? (These experiments are missing in my opinion.)

[c] https://arxiv.org/abs/2110.09192

- Also, how would the ImageNet results change when performing trials instead of fixing calibration and test set?
- In any of the non-ImageNet experiments, are there cases where baselines do not provide calibration set/threshold-conditional coverage and could this be evaluated somehow?

Conclusion:
I believe the proposed method is a good contribution to the community. I only see two drawbacks: First, the paper is very difficult to follow, requiring the reader to sit down and be able to annotate the paper and write down definitions/notation to follow the algorithm and discussion. I think this could be improved by devoting more space to it and maybe moving some experiments or discussion of Gupta et al. to the appendix. Second, for some experiments the picture is less clear and this should be discussed a bit better – i.e., in which settings the proposed method really works well/better. This also includes evaluating or commenting the threshold-conditional coverage which is “advertised” in the introduction. While I was unable to check the proof in the appendix in detail, due to density and amount of experiments, the proof sketch seems reasonable and valid as far as I can tell, but it would be good if another review could confirm this.

---

> ### Author Response · Authors · 2022-08-02
> **Response to Reviewer RP3U Part 1**
>
>
> Thank you so much for the effort that you put into your detailed review! We will take your expositional points to heart and incorporate your suggestions to improve clarity.
>
> **Relationship to Gupta et al.:** The most important improvement compared to Gupta et al. is this: Their algorithm for obtaining multivalid coverage involved, at every round, calling the Ellipsoid algorithm with a greedy separation oracle to solve a linear program with infinitely many constraints and hence has no practical implementation (and they do not implement it). We give a very simple combinatorial algorithm for the same problem, give a fast implementation, and run a set of experimental comparisons. We also give various improvements related to the convergence rates, but we appreciate that the discussion of some of these could be moved to the supplement to avoid obscuring the main point.
>
> **Comparison to [a] and [b]:** We note that [a] and [b] both study a very different kind of conditional guarantee --- validity conditional on the training/calibration set. They do not provide the kind of conditional coverage guarantee that we provide, which conditions on both the group membership of a point at test time as well as on the threshold used at test time.  (As an aside, in the online setup, there is no distinction between the training and test set; instead, we provide the stronger guarantee of coverage for every realized transcript.) We will add detailed discussions of [a] and [b] to our related work.
>
> More generally, we note that split conformal prediction (and all prior work we are aware of) is currently unable to provide the kind of tight group-conditional coverage guarantees we have when the groups intersect, even under the assumption that the data is exchangeable. Two prior papers (which we cite) have studied group conditional coverage in the exchangeable setting: Romano et al. (2020) and Barber et al. (2020). Romano et al. consider separately calibrating on each group, but this method only applies to disjoint groups. Barber et al. study intersecting groups and separately calibrate on each one, and then at test time, use the most conservative threshold amongst all groups that a test example $x$ is a member of. This method over-covers --- i.e. it does not converge to the target coverage level even in the limit (and does not offer threshold conditional coverage).
>
> Questions about the plots/experiments:
>
> **Figure 2, difficulty of groups**: Indeed, in this synthetic data experiment, although we have 20 overlapping groups, by construction, there is (unknown to the algorithm) one low noise group (group 0) and one high noise group (group 1). The other groups intersect with these but are not themselves noise-relevant. Although it is simple, we think this is an interesting experiment because the algorithm must discover which groups are noise relevant. And despite it's simplicity, this setup is  already enough to demonstrate that prior methods do not achieve tight group conditional coverage.
> In Figure 3, we repeat the same experiment on real census derived data, with groups defined by race and gender. As you predict, coverage levels vary in more interesting (although less pronounced) ways across these groups.
>
> **Adding noise in time series experiments:** We chose the stock prediction dataset so that we can do a direct comparison to ACI on the dataset they use in their paper.
> We add noise to make the behavior of the different subgroups (i.e., different days of the week) we consider interesting.
> In particular, financial data does not exhibit noticeably different statistical properties on any simple groupings of days (or else this would be exploitable). We add noise to different subsets of the days to produce an experiment in which the uncertainty of the model is quantifiably different within different groups. We will add a sentence to clarify why our subgroup experiment is synthetic in the revision.
>
> **Figure 3**: We compare to two variants of split conformal prediction. The first variant ignores the group structure, and hence uses the same threshold on all examples. The second (the method of Barber et al. 2020) calibrates a threshold for each group, and then uses the most conservative threshold amongst all groups an example is a member of --- this one uses different thresholds for different examples.

---

> > ### Comment · Reviewer_RP3U · 2022-08-07
> > **Thanks for the detailed rebuttal**
> >
> > I appreciate the detailed rebuttal and clarifications.

---

> ### Author Response · Authors · 2022-08-02
> **Response to Reviewer RP3U Part 2**
>
>
> **Figure 4a, undercoverage:** ACI undercovers on all the displayed groups other than the entire sequence (on which it achieves exact marginal coverage), but it actually significantly over-covers on the complement of the union of these groups (not shown in the plot), thus compensating for the displayed under-coverage on the groups of interest. We will add a sentence clarifying this point in the revision.
>
> **ImageNet:** Re the calibration set size we use for RAPS: We experimented with larger calibration sets, but they did not affect the results significantly. Since RAPS is a split conformal method, it uses a single threshold and so is threshold calibrated, but because of this cannot offer group conditional guarantees. We did not repeat our group conditional and distribution shift experiments on the ImageNet data, in part because our paper including our experimental evaluation was already very long, but you are right that adding these would make the experimental section more comprehensive, and we will plan to do so.
>
> **Threshold Conditional Coverage:** Yes, in our time series experiments with increasing nonconformity scores, we show that ACI alternates between full coverage and under-coverage, which shows a lack of threshold calibrated coverage (since, e.g., conditional on predicting the trivial prediction set/threshold, the coverage must be 100\%).
>
> Thanks once again for the detailed suggestions in your review --- these will be very helpful as we edit for clarity in the revision!

---

### Official Review · Reviewer_3pUm · 2022-07-12

**Rating:** 7
**Confidence:** 3
**Soundness:** 3 good
**Presentation:** 3 good
**Contribution:** 4 excellent

**Summary:**

This paper presents an online conformal prediction algorithm, which is proven to satisfy a “threshold-calibrated multivalid coverage guarantee” (Definition 2.1). Compared with prior work, several advantages of the proposed algorithm are: 1) it does not require the exchangeability assumption for proving the coverage guarantee, thus it can be used to model sequential data such as time series; 2) it is claimed to be robust to arbitrary and unanticipated distribution shift; 3) the obtained coverage guarantee is not just marginally but promises group-conditional coverage; 4) it achieves improved statistical rate over the work of [Gupta et al., 2022]. Experiments on synthetic data, time-series data and image data support these advantages over existing methods.


**Questions:**

1. In the experiments, most of the considered tasks are regression tasks. You only compared with Angelopoulos et al. [2020] on marginal coverage, how about the conditional coverage for subgroups? In general, I am curious about the conditional coverage performance of your algorithm for classification tasks.

2. How do you decide the number of buckets m for your algorithm for a practical task?

**Limitations:**

Yes

**Strengths And Weaknesses:**

Strength:

1. The paper is well-motivated, and its contribution is significant to the field of conformal prediction. The proposed method achieves both worst-case empirical coverage and calibrated, multivalid coverage, which clearly advances the existing literature.

2. The paper discusses the comparisons with related works well, and is technically solid with clearly presented definitions, assumptions, theorems, and proofs.

3. The proposed surrogate loss (Definition 3.4) and its induced techniques for proving the threshold-calibrated multivalid coverage is theoretically interesting.


Weakness:

1. Section 2 and Section 3 are overloaded with technical notations. The motivations of the proposed coverage guarantee (Definition 2.1) and the specific design of Algorithm 1 are not well-explained, at least intuitively. For example, Definition 2.1 introduces threshold calibration, you may want to explain why you want the marginal coverage to be additionally satisfied for multiple buckets simultaneously, as opposed to the case where there is just a single bucket.

2. In line 32, the authors claimed that the proposed method has worse-case adversarial guarantees, which is very strong in my perspective. However, the threat model is not defined clearly (without a formal definition) at the beginning, thus hard to evaluate its significance. What are the differences between the adversarial setting considered here and the setting of covariate shift such as Tibshirani et al. [2019]? Does the adversary need to satisfy some assumptions (e.g., Definition 3.1)? [1] is a related paper on this, so should be mentioned or discussed in the paper as well.

[1] Adversarially Robust Conformal Prediction, Asaf Gendler and Tsui-Wei Weng and Luca Daniel and Yaniv Romano, ICLR 2022


Typos:

Line 148: n is not defined

Line 149: w belongs to -> q belongs to

---

> ### Author Response · Authors · 2022-08-01
> **Response to Reviewer 3pUM**
>
> Thank you for your review! We take to heart your point that our paper is heavy with notation, and will look for ways to reduce the notational burden while maintaining rigor. Below we answer your questions:
>
> **Strength of our adversary:** The adversary in the online adversarial model is indeed very strong, and the fact that we can get positive results in this model means that our algorithm is very robust. Specifically, the adversary can choose any covariates $x$ and any distribution over labels $y$ at each step; the only constraint being that the induced distributions over non-conformity scores  are ``smooth'', which is satisfied (with some parameters) whenever the distribution on non-conformity scores is continuous. Assumptions like this are always necessary when the goal is not just to get conservative coverage, but to converge to the target coverage rate exactly. A common technique in the literature to enforce this assumption if it does not hold in the data naturally is to perturb non-conformity scores with arbitrarily small amounts of continuous noise.
>
> **Comparsion to Tibshirani et al. [2019]:** Our assumption is substantially weaker than the covariate shift setting of Tibshirani et al [2019], which assumes that the data shifts a single time in a known and proscribed way between calibration and test: in particular, that the feature distribution changes via a reweighting using known propensity scores, and that the conditional label distribution remains unchanged. In the online setting we consider, there is no division drawn between calibration and test, and the feature distribution can change in arbitrary ways at every round (that need not be via known propensity scores), as can the conditional label distribution.
>
> **Comparison to [1]**: [1] studies "adversarially perturbed data" in the spirit of the "adversarial examples" literature. They assume that the data is exchangeable, but that the test data is perturbed by an adversary who is limited to modifying each example's features within a ball of small norm. Their techniques are very different from ours (and much closer to traditional split conformal prediction), and leverage the fact that the underlying distribution (except for the perturbations) is exchangeable, which we do not require. We will add a detailed discussion of [1] to our related work.
>
> **Conditional coverage on classification tasks:** Our ImageNet experiments demonstrate that our techniques can be applied to arbitrary non-conformity scores, including those recently developed for classification settings. On this experiment, we did not repeat our earlier investigations of subgroup and threshold conditional coverage and distribution shift since we investigate these issues extensively in our other experiments (and our paper is already quite long).
>
>
> Our actual algorithm operates directly on the nonconformity scores and is agnostic to the process that generates them (in particular, if the task is classification or regression), so we have no reason to expect the results to be different. Nevertheless, you are right that this would be a natural experiment that would make our collection of results more comprehensive, and we will consider adding it.
>
> **Number of calibration buckets:** In all of our experiments we choose m = 40 calibration buckets. This isn't a principled choice (i.e. we did not do any hyper-parameter optimization on a task by task basis) --- just a simple choice that seemed to work well consistently. With more careful hyper-parameter tuning our results would presumably improve slightly.

---

> > ### Comment · Reviewer_3pUm · 2022-08-08
> > **Thanks for the rebuttal**
> >
> > Thank you for the clarifications! My concerns are resolved. I recommend acceptance.

---

> > > ### Author Response · Authors · 2022-08-08
> > > **Thanks**
> > >
> > > Thank you --- and thanks once again for the time you spent reviewing!

---

### Meta-Review · Area_Chair_8xPS · 2022-08-27

**Recommendation:** Accept
**Confidence:** Certain

**Metareview:**

This paper proposes a conformal prediction based method for sequential prediction, relaxing the exchangeability assumption.  It is robust to distribution shift, and achieves group-conditional coverage guarantees.  The method is efficient, novel, and outperforms existing methods.  All the reviewers, including myself, find the paper a solid contribution to the methodology and analysis, hence a clear accept.

**Award:**

No

---

### Decision · Program_Chairs · 2022-09-14

Accept